# KLF-1 orchestrates a xenobiotic detoxification program essential for longevity of mitochondrial mutants

Marija Herholz[1], Estela Cepeda[1], Linda Baumann[1], Alexandra Kukat[1], Johannes Hermeling[1], Sarah Maciej[1], Karolina Szczepanowska[1], Victor Pavlenko[1], Peter Frommolt[1] & Aleksandra Trifunovic [1,2]

Most manipulations that extend lifespan also increase resistance to various stress factors and environmental cues in a range of animals from yeast to mammals. However, the underlying molecular mechanisms regulating stress resistance during aging are still largely unknown. Here we identify Krüppel-like factor 1 (KLF-1) as a mediator of a cytoprotective response that dictates longevity induced by reduced mitochondrial function. A redox-regulated KLF-1 activation and transfer to the nucleus coincides with the peak of somatic mitochondrial biogenesis that occurs around a transition from larval stage L3 to D1. We further show that KLF-1 activates genes involved in the xenobiotic detoxification programme and identified cytochrome P450 oxidases, the KLF-1 main effectors, as longevity-assurance factors of mitochondrial mutants. Collectively, these findings underline the importance of the xenobiotic detoxification in the mitohormetic, longevity assurance pathway and identify KLF-1 as a central factor in orchestrating this response.

---

[1] Cologne Excellence Cluster on Cellular Stress Responses in Ageing-Associated Diseases (CECAD) and Institute for Mitochondrial Diseases and Ageing, Medical Faculty, University of Cologne, D-50931 Cologne, Germany. [2] Center for Molecular Medicine Cologne (CMMC), D-50931 Cologne, Germany. Correspondence and requests for materials should be addressed to A.T. (email: aleksandra.trifunovic@uk-koeln.de)

Lifespan extension in different species can be achieved by various genetic manipulations and treatments, such as disruption of insulin/IGF1 signalling, decrease in mitochondrial respiration, suppression of translation or caloric restriction. Despite very different origins of these longevity programmes, they all warrant increased resistance to various stresses like heat, oxidative stress or radiation[1,2]. Although the concept that lifespan might depend on the capacity to withstand external stress cues is very old, little is currently known about signalling pathways underlying these cytoprotective responses and their ability to affect lifespan. Furthermore, how much an individual cytoprotective mechanism contributes to the lifespan extension induced by different manipulations is a key question that remains to be answered[2].

Transcription profiling of many long-lived mutants from worm to mouse has recently revealed that upregulation of a number of genes involved in xenobiotic detoxification is common to longevity-assurance pathways across different phyla[1]. Xenobiotic detoxification includes activation of drug-metabolizing enzymes (DMEs), which are classified in two main groups: phase I—mainly cytochrome P450 oxidases (CYPs) and phase II—mainly UDP-glucuronosyltransferases (UGTs), glutathione-S-transferases (GSTs), sulfotransferases, and acetyltransferases, coupled to the activity of phase III transporters that mediate the efflux of metabolic end products out of the cells after the completion of phase II conjugation[3].

Interestingly, analyses of expression profiles from long-lived mice, including calorically restricted mice, different dwarf mice or mice treated with rapamycin, revealed that many CYPs are upregulated and positively correlate with increased longevity[4,5]. Moreover, increased expression of multiple *cyp* genes was reported in diverse long-lived *C. elegans* models, including mitochondrial mutants[6,7]. Although interesting, these findings provided just a correlative connection to longevity. Importantly, little is known about the activation and regulation of xenobiotic responses in respect to longevity, since most studies to date have focused only on describing changes in the DME expression in different mutants and conditions[1].

Here we identify Krüppel-like factor 1 (KLF-1) as a major regulator of the detoxification response involved in longevity assurance in *Caenorhabditis elegans*. We show that upon mild mitochondrial dysfunction and/or oxidative stress, KLF-1 translocates to the nucleus and activates *cyp* genes that in different organisms often encode enzymes involved in the xenobiotic detoxification process. We further show that a redox-regulated KLF-1 activation coincides with the peak of somatic mitochondrial biogenesis that occurs between L3 and D1 and is essential for the mitohormetic response that dictates longevity.

## Results

### KLF-1 regulates the mitomutant longevity during adulthood.
We performed a genome-wide RNAi screen starting with chromosome III of the existing feeding library[8] in *isp-1(qm150);ctb-1 (qm189)*, a long-lived mitochondrial mutant that carries two mutations in complex III subunits[9]. We focused ultimately on transcription factors that suppress longevity upon knockdown, but have no effect on wild-type (WT) lifespan. Only *klf-1* fulfilled these criteria (Fig. 1a). KLF-1 depletion in the *isp-1(qm150)* single mutant or the short-lived mitochondrial mutants, *gas-1(fc21)* and *mev-1(kn1)*, also led to a lifespan decrease (Supplementary Fig. 1a–c) in agreement with our previous observation that both long- and short-lived mutants activate the same longevity assurance responses[10].

Mitochondrial dysfunction needs to occur during late *C. elegans* developmental stages to assure the longevity phenotype[11].

Remarkably, *klf-1* knockdown only during adulthood (Fig. 1b) completely suppressed the long-lived phenotype in the *isp-1;ctb-1* mutant, whereas knockdown during developmental stages did not have an effect (Fig. 1c), even when performed over two generation (Supplementary Fig. S1d). Similarly, *klf-1* deficiency during adulthood decreased the longevity induced by the developmental knockdown of different mitochondrial respiratory chain (MRC) subunits (Supplementary Fig. 1e). KLF-1 activity in the early adulthood (first day of adulthood—D1) seems to be crucial for longevity assurance, since the *klf-1* deficiency starting from third or fifth day of adulthood, had much milder effect on the phenotype (Supplementary Fig. 1f). This is further complemented with the observed 2.5-fold increase in the *klf-1* expression in *isp-1; ctb-1* mutant at D1, levels that are reached in WT animals only later in the adulthood (Fig. 1d, e).

### KLF-1 does not act through UPR^mt or HIF-1 pathway.
*C. elegans klf-1* encodes a protein of the Krüppel-like transcription factor family (KLF) that shares high homology to the Zn-finger domain of other KLF proteins (Supplementary Fig. 2a) and its closest mammalian homologues are KLF2, KLF4 and KLF5. In worms, *klf-1* is expressed in the intestine, hypodermis and a handful of neurons (Supplementary Fig. 2b). Recently, KLF-1 has been implicated in the regulation of the longevity phenotype upon dietary restriction and insulin signalling, together with KLF-3, through its role in the regulation of autophagy[12,13]. However, the exact function of KLF-1 and its downstream targets remains largely unknown.

To understand the role of KLF-1 in mitomutants longevity, we analysed it in the context of the previously reported *isp-1;ctb-1* phenotypes[9]. While KLF-1 depletion did not affect the delayed development and the smaller brood size of the *isp-1;ctb-1*, the lower movement rates (at the fifth day of adulthood—D5) were further decreased (Supplementary Fig. 3a–c). Likewise, *klf-1* knockdown at D5 caused a further decrease in the respiration rate of *isp-1;ctb-1* (Supplementary Fig. 3d). Nevertheless, this decrease is unlikely to contribute to the loss-of-longevity phenotype since a much stronger drop in oxygen consumption after *cco-1* knockdown, a subunit of respiratory complex IV, in the *isp-1;ctb-1* animals further prolongs lifespan (Supplementary Fig. 3e).

To further elucidate the KLF-1 role in longevity assurance of *isp-1;ctb-1* mutants, we analysed pathways previously shown to be important for lifespan extension by mitochondrial dysfunction, like the mitochondrial unfolded protein response (UPR^mt) and HIF-1 pathway[11,14]. Loss of KLF-1, however, did not show any significant effect on the highly activated UPR^mt in *isp-1;ctb-1* animals (Fig. 2a, b). Similarly, our results imply that the HIF-1 pathway is not directly regulated by KLF-1 as we did not detect a change in the level of its *bona fide* targets (*nhr-57* and *egl-9*) in young *isp-1;ctb-1* mutants (Fig. 2c). Remarkably, the HIF-1 pathway seems to be suppressed in *isp-1;ctb-1* mutants at D5 of adulthood, a feature that is lost upon KLF-1 depletion (Fig. 2c). Thus, our findings strongly imply that (i) both the UPR^mt and HIF-1 pathways are only indirectly affected by the hormetic response later in life (D5) by the KLF-1 depletion; (ii) although these pathways could still contribute to the observed phenotypes, they are most likely not the main source of lifespan reduction caused by the *klf-1* depletion.

### KLF-1 is vital for a response to the oxidative stress.
Increased longevity in mitochondrial mutants has been correlated to the increased resistance to oxidative stress and ROS-mediated signalling[15,16]. Hence, we investigated the antioxidant capacity of the *isp-1;ctb-1* mutant by exposing them chronically (from L4) to the ROS-producing reagent paraquat. As expected, *isp-1;ctb-1*

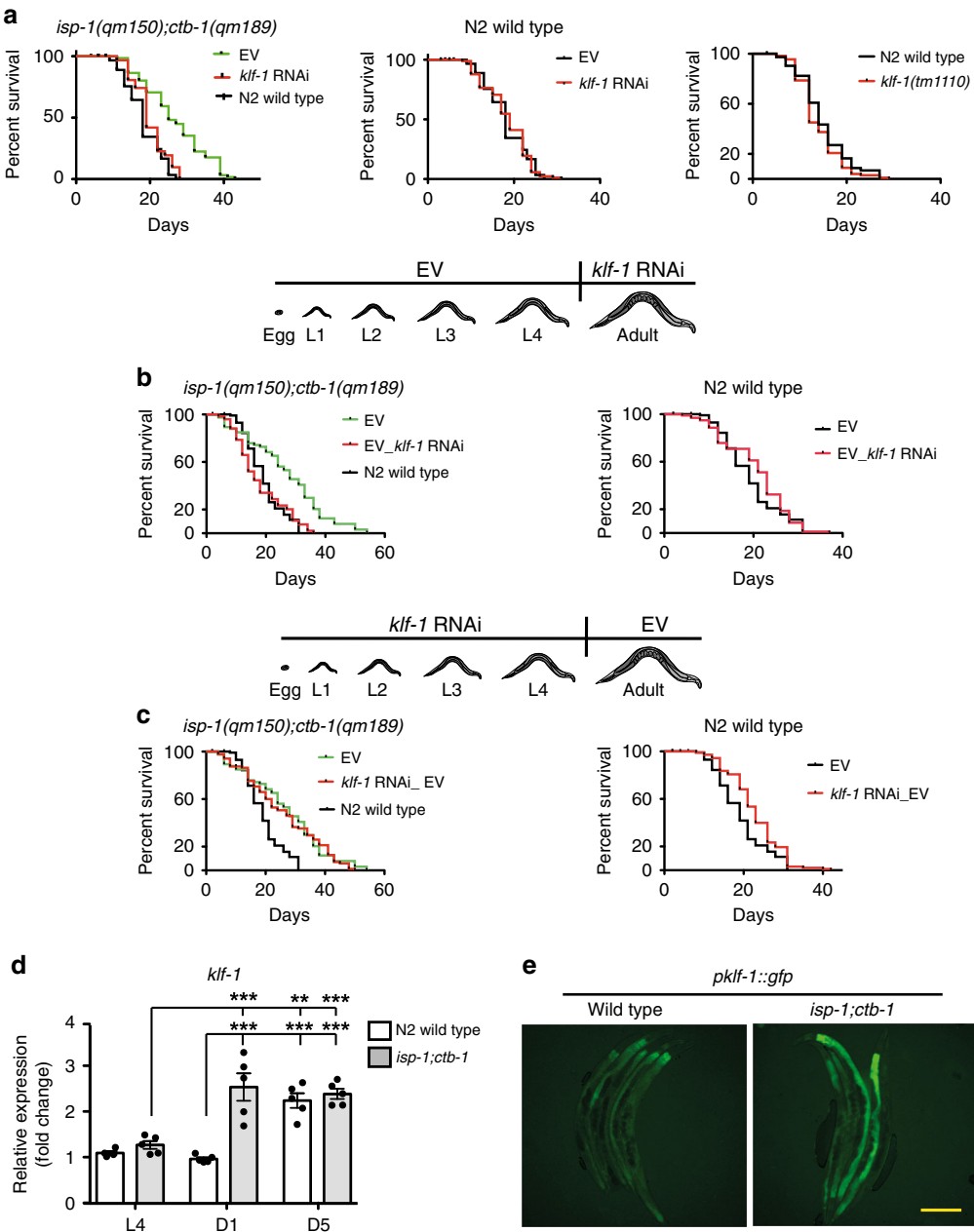

**Fig. 1** KLF-1 mediates the longevity of mitochondrial *isp-1;ctb-1* mutant. **a** Lifespan curves of *isp-1(qm150);ctb-1(qm189)* mutant (left) and wild-type animals (middle) grown on control (empty vector, EV) or *klf-1* RNAi plates. Right panel shows lifespan curve of *klf-1(tm1110)* mutant animals. **b** *isp-1(qm150);ctb-1(qm189)* mutants (left) and wild type animals (right) were grown on control (EV) RNAi plates until L4 developmental stage and then transferred to *klf-1* RNAi. **c** Lifespan curve of *isp-1(qm150);ctb-1(qm189)* mutant (left) and wild-type animals (right) grown on *klf-1* RNAi plates until L4 developmental stage and then transferred to control (EV) RNAi. **d** *klf-1* expression level was assessed by qPCR at the L4 larval stage, the first (D1) and the fifth day (D5) of adulthood, in the wild-type and *isp-1(qm150);ctb-1(qm189)* mutants. Data are presented as mean ± SEM. **p < 0.01, ***p < 0.001, one-way ANOVA with Tukey post hoc test. n = 5 samples per condition. **e** Fluorescent images of *gfp* expression under *klf-1* promoter in wild type and *isp-1(qm150);ctb-1(qm189)* at the first day of adulthood. Scale bar 100 μm

showed increased resistance to paraquat across all different life-stages (Supplementary Fig. 4a). The increased resistance of older *isp-1;ctb-1* animals (D5) seemed to be dependent on KLF-1. Likewise, *klf-1* expression was specifically upregulated by the high paraquat treatment, and no other exogenous stressors, such as heat shock or osmotic stress (Fig. 3a).

Increased resistance to oxidative stress has been attributed to a mitohormetic response, where early exposure to stress, in this case ROS, elicits cytoprotective mechanisms that render animals more resistant to the same stress factors later in life[17]. If hormetic responses were active in the *isp-1;ctb-1*, one would expect that through adulthood the ROS-induced damage would not rise. Indeed, the amount of carbonylated proteins as a measure of oxidative damage, increased with age in the WT worms, whereas in the *isp-1;ctb-1* they were mainly unchanged (Fig. 3b, c). In some cases, we observed the opposing effect where the oxidative damage in *isp-1;ctb-1* mutants that was high at the D1, would decrease in the later stages (D5), to match the level of young adult WT controls (Fig. 3b). The increase in oxidative stress at the D1 in *isp-1;ctb-1* was accompanied by upregulation of SOD-2/

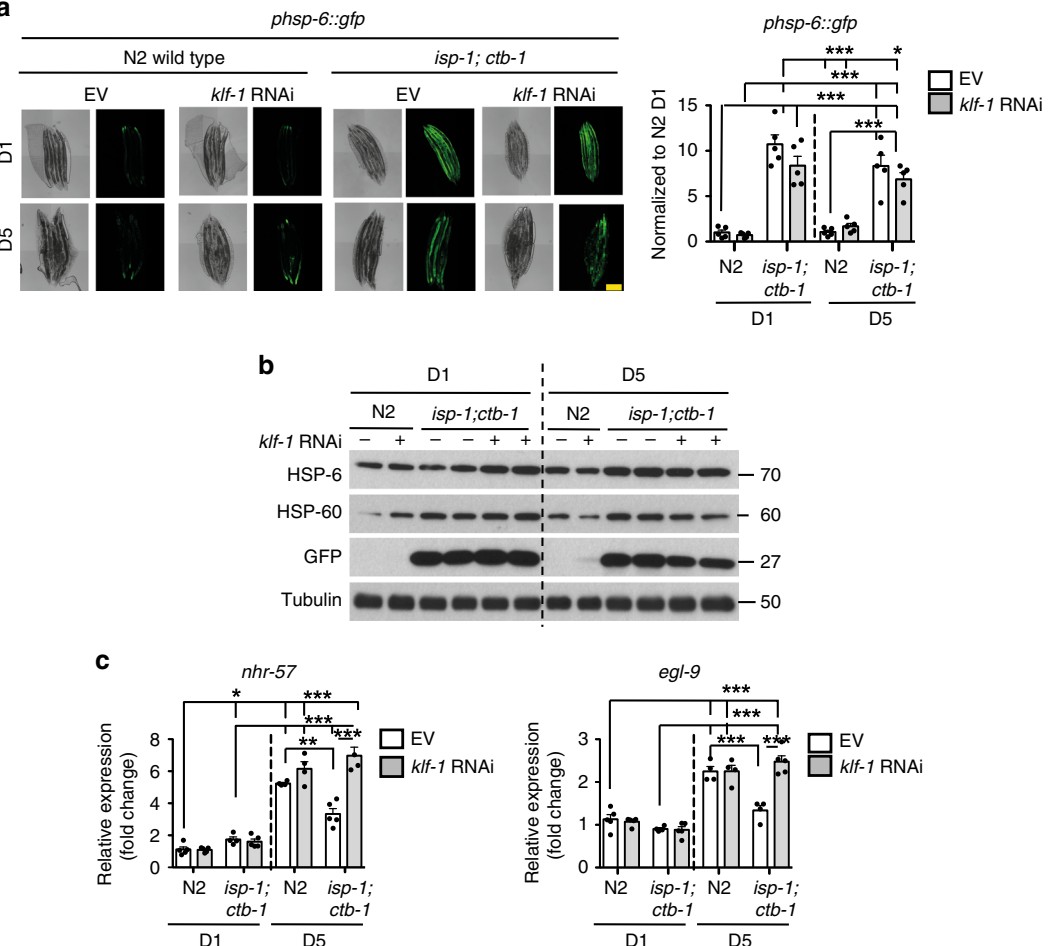

**Fig. 2** KLF-1 mildly affects the mitochondrial UPR and does not directly affect the HIF-1 pathway. **a** Mitochondrial UPR was assayed by activation of *gfp* expression under the control of *hsp-6* promoter. Confocal images (left) were taken at the first (D1) and the fifth (D5) day of adulthood in wild type and *isp-1 (qm150);ctb-1(qm189)* mutant grown on control (EV) or *klf-1* RNAi. Scale bar 100 μm. Quantification is shown on the right. Data are presented as mean ± SEM. *$p < 0.05$, ***$p < 0.001$, one-way ANOVA with Tukey post hoc test. $n = 5$ animals per condition. **b** Western blots showing levels of HSP-6, HSP-60 and GFP in proteins isolated from *phsp-6::gfp* transgenic strain, in N2 wild-type and *isp-1(qm150);ctb-1(qm189)* genetic background at D1 and D5 of adulthood. Tubulin was used as a loading control. **c** Targets of the HIF-1 transcription factor, *nhr-57* and *egl-9* were assayed for their mRNA expression on D1 and D5 of adulthood in N2 wild-type and *isp-1(qm150);ctb-1(qm189)* mutant. Data are presented as mean ± SEM. *$p < 0.05$, **$p < 0.01$, ***$p < 0.001$, one-way ANOVA with Tukey post hoc test. $n = 4$ independent samples per condition

MnSOD, a major mitochondrial antioxidant enzyme that takes care of superoxide ($O^{2-}$) (Fig. 3b, d). Curiously, KLF-1 depletion prevented the early upregulation of SOD-2/MnSOD, at both transcript and protein level, but had no effect on them later in the adulthood (D5), suggesting that *sod-2*, is not directly regulated by KLF-1 (Fig. 3b, d and Supplementary Fig. 4b). Likewise, we show that KLF-1 does not regulate transcription of *sod-1*, a main cytosolic SOD or *sod-3*, a second mitochondrial MnSOD that is controlled by DAF-16 transcription factor and highly induced in *daf-2* long-lived mutants (Supplementary Fig. 4b, c). KLF-1 is also not involved in the transcription regulation of catalases, which are major $H_2O_2$ metabolizing enzymes (Supplementary Fig. 4b). Collectively, these data strongly suggest that *isp-1;ctb-1* mutants activate pathways, other than bona fide antioxidant response, to combat increased oxidative stress imposed by mitochondrial dysfunction. These responses seem to depend on KLF-1 activity that regulates the hormetic response observed in mitochondrial mutants.

**KLF-1 regulates phase I detoxification response genes**. To elucidate the cytoprotective machinery, which mediates the

hormetic response in the *isp-1;ctb-1* mutant, we compared the transcriptome of the D5 *isp-1;ctb-1* animals with the same animals grown either on *klf-1* RNAi during development (long-lived phenotype) or *klf-1* RNAi during adulthood (suppressed longevity). These findings were then matched to those of WT. In total, we investigated 251 genes, whose expression was altered in *isp-1;ctb-1*, but normalized to WT levels upon *klf-1* knockdown during adulthood (Table 1 and Supplementary Data 1). In agreement with the previous results[15,16], the list of genes upregulated in the mitochondrial mutant did not show significant enrichment for genes involved in the ROS detoxification or damage repair. Remarkably, we also did not observe changes in genes encoding mitochondrial proteins, including OXPHOS subunits. The DAVID gene enrichment analysis tool was used to evaluate the statistical representation of gene categories as defined by gene ontology[18]. This approach identified "oxidation–reduction" to be the most affected biological process, and "metabolism of xenobiotics by cytochrome P450" and "drug metabolism" to be the two most affected KEGG pathways in *isp-1;ctb-1* mutant upon lifespan-shortening *klf-1* knockdown (Supplementary Tables 1 and 2). Correspondingly, in the mitochondrial mutant there was a

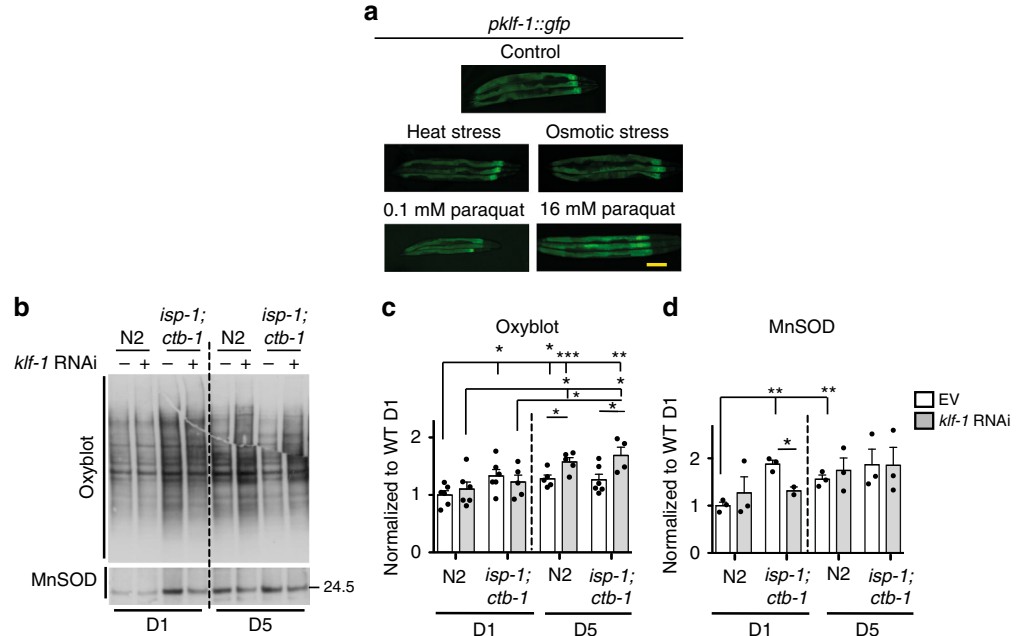

**Fig. 3** *klf-1* expression increases specifically upon oxidative stress, but not other types of stresses. **a** Fluorescent images of a strain expressing *gfp* under the *klf-1* promoter. Young adults were exposed to osmotic and heat stress or 16 mM paraquat. For 0.1 mM paraquat, animals were treated with the drug during the larval development and imaged at the first day of adulthood. Scale bar 100 μm. **b** Oxyblot was performed on proteins isolated from the first (D1) or the fifth (D5) day old N2 wild-type and *isp-1(qm150);ctb-1(qm189)* animals. The bottom panel shows the same oxyblot membrane probed against SOD-2/MnSOD antibody. **c** Quantification of the oxyblot membranes of D1 and D5 old animals. Data are normalized to N2 D1 levels. Ponceau staining was used as loading control. Data are presented as mean ± SEM. *$p < 0.05$, **$p < 0.01$, ***$p < 0.01$, one-way ANOVA with Tukey post hoc test. $n = 5$ independent samples per condition. **d** Quantification of SOD-2/MnSOD levels of D1 and D5 old animals. Data are normalized to N2 D1 levels. Ponceau staining was used as loading control. Data are presented as mean ± SEM. *$p < 0.05$, **$p < 0.01$, one-way ANOVA with Tukey post hoc test. $n = 3$ independent samples per condition

| **Table 1 Microarray data available on NCBI Gene Expression Omnibus (GEO)** | |
|---|---|
| **GEO accession number** | **GSE61771** |
| **GSM Acc** | **Description** |
| GSM1513691 GSM1513692 GSM1513693 | D5 *isp-1(qm150);ctb-1(qm189)* grown on control L4440 RNAi and switched to *klf-1* RNAi at L4 stage |
| GSM1513694 GSM1513695 GSM1513696 | D5 *isp-1(qm150);ctb-1(qm189)* grown on *klf-1* RNAi and switched to control L4440 RNAi at L4 stage |
| GSM1513697 GSM1513698 GSM1513699 | D5 *isp-1(qm150);ctb-1(qm189)* grown on control L4440 RNAi |
| GSM1513700 GSM1513701 GSM1513702 | D5 N2 wild type grown on control L4440 RNAi |

significant overrepresentation of genes encoding proteins involved in phase I xenobiotic detoxification, particularly cyto-chrome P450 oxidases (CYPs) (Fig. 4a). In the *isp-1;ctb-1* mutant, most of the identified *cyps* showed highly increased, KLF-1-dependent expression early in life, which often declined during aging (Fig. 4b). In agreement, promoter analyses of most significantly changed genes, including majority of identified *cyps*, revealed one or more consensus KLF-binding elements (CA/GCCC) within 2000 bp upstream and 100 bp downstream of the transcription start site (Supplementary Data 2).

To identify direct transcriptional targets of KLF-1 involved in the regulation of detoxification response, we performed chromatin immunoprecipitation followed by DNA sequencing (ChIP-seq) analyses in WT overexpressing KLF-1-YFP protein or *isp-1;ctb-1* mutants (Supplementary Data 3). The analysis identified a number of genomic regions (on average 430 and 550, respectively), out of which 149 were found in both data sets (Supplementary Data 3). We identified a KLF-1-binding region immediately upstream of the four *cyp* genes, including *cyp-13A11*, one of the most upregulated genes in our microarray data set (Supplementary Data 3). Further qPCR analysis on ChIP samples demonstrated that, indeed KLF-1 is present on *cyp-13A11* promotor region in WT, but it binds the same region in *isp-1; ctb-1* mutants more efficiently (Fig. 4c). Remarkably, the ChIP-Seq data not only strongly overlapped with our microarray results (Supplementary Tables 1 and 2 and Supplementary Data 3), but more than 80% of detected KLF-1 targets were in genes previously shown to be responsive to either, treatment with oxidative stress-producing agents (paraquat or rotenone)[19], or agents that block mtDNA synthesis and therefore lead to strong mitochondrial OXPHOS dysfunction (EtBr or nucleoside reverse transcriptase inhibitors—NRTIs)[20,21]. In most cases KLF-1 target genes were found in more than one data set (Supplementary Data 3). Therefore, together with previous microarray and promoter analysis, ChIP-Seq data strongly support our hypothesis that KLF-1 directly regulates at least some of the *cyps*.

Two classes of CYPs whose expression was most highly upregulated in the *isp-1;ctb-1* in a KLF-1-dependent manner are homologues of murine CYP2C70 and CYP3A13. To test whether the induction of *cyp* expression upon mitochondrial dysfunction is conserved, we treated Hepa1-6 cells with low level of antimycin

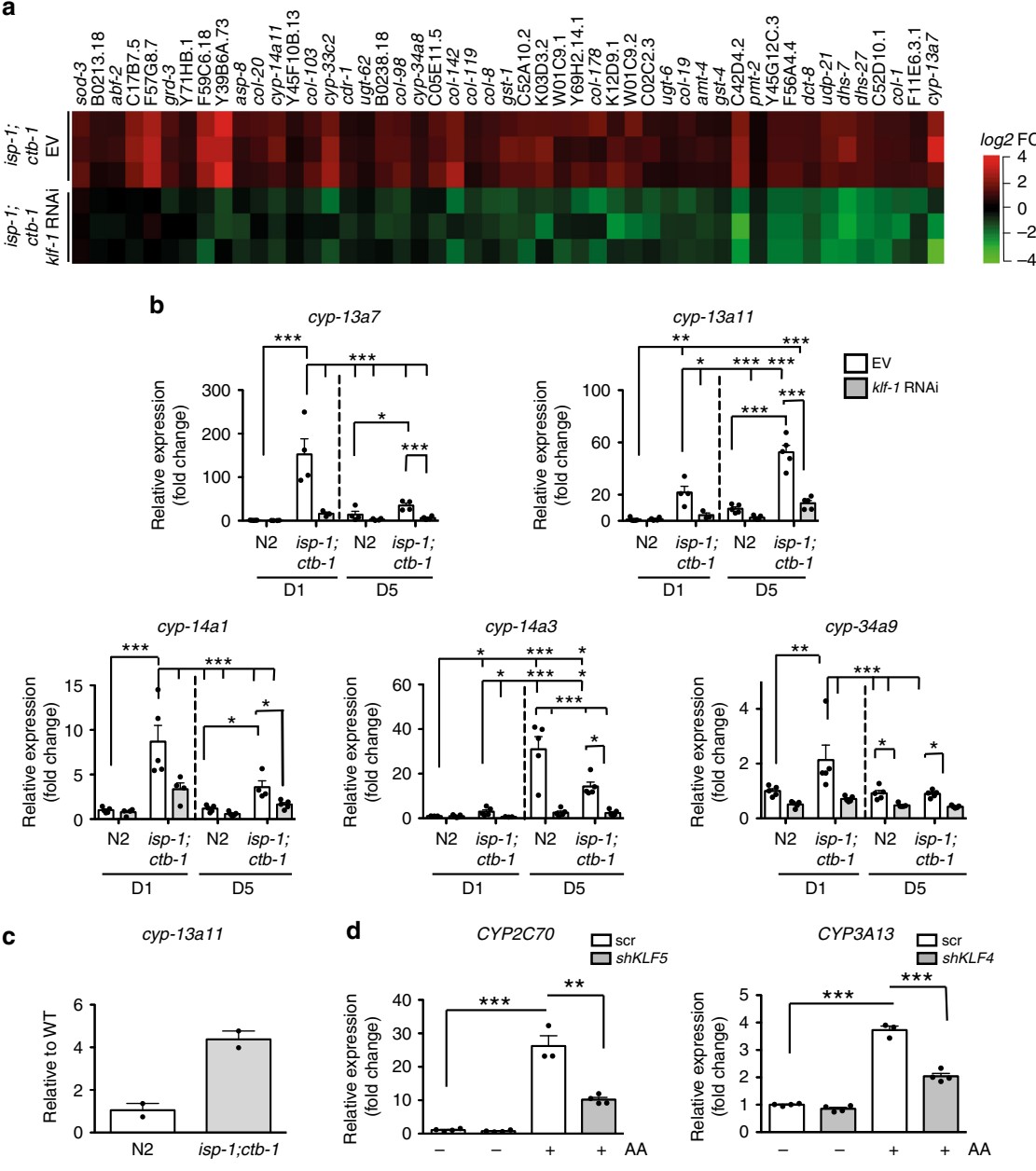

**Fig. 4** KLF-1 mediates expression of cytochrome P450 oxidases. **a** log2 fold-change in gene expression in *isp-1(qm150);ctb-1(qm189)* mutant animals compared with N2 wild type, grown either on control (EV) or *klf-1* RNAi from L4 larval stage. Microarray analysis was performed at the fifth day of adulthood. **b** The expression of cytochrome P450 oxidases from (**a**) was confirmed by qPCR. *n* = 4 independent samples per condition. Animals were analysed at the first (D1) or the fifth (D5) day of adulthood and grown on control (EV) and *klf-1* RNAi whole life. **c** CHIP-qPCR analysis of WT and *isp-1;ctb-1* for *cyp-13a11* promoter region. *n* = 2 independent replicates. **d** Levels of *Cyp2C70* and *Cyp3A13* were assayed with qPCR in Hepa1-6 cells with or without antimycin A (AA) treatment as indicated. Cells were exposed to either *Klf4*, *Klf5* or control siRNA. Data are presented as mean ± SEM. *p < 0.05, **p < 0.01, ***p < 0.001, one-way ANOVA with Tukey post hoc test. *n* = 4 independent samples per condition

A (a complex III inhibitor and a potent ROS inducer) and detected high increase in *Cyp2C70* and *Cyp3A13*, supporting the notion that the response to mitochondrial dysfunction is similar in *C. elegans* and mammals (Fig. 4d). The upregulation of these genes was dependent on the closest mammalian homologues of KLF-1, namely KLF4 and KLF5, providing further evidence for the high conservation of this stress response (Fig. 4d).

The activation of *cyp* genes was exclusively driven by KLF-1 and not the other two *C. elegans* KLF homologues, KLF-2 and KLF-3 (Supplementary Fig. 5a). As it was recently reported that KLF-3, together with KLF-1, plays an important role in the regulation of autophagy genes necessary for caloric restriction and insulin signalling mediated longevity[13], we further evaluated possible role of KLF-3 in regulating *cyp* genes. However, our results argue against the role of KLF-3 in this process, as its depletion did not affect the expression of *cyp-25a2*, while the combined *klf-1* and *klf-3* knockdown produced intermittent effect, possibly due to dilution of individual RNAi (Supplementary Fig. 5b). We also show that in contrast to *klf-1*, the *klf-2* and *klf-3* expression was not affected in the *isp-1;ctb-1* (Supplementary Fig. 5c). However, the loss of KLF-3 strongly suppressed the *isp-1; ctb-1* lifespan suggesting an important role for this

transcription factor in the context of mitochondrial dysfunction, outside of regulation of phase I detoxification (Supplementary Fig. 5d). Remarkably, combined depletion of *klf-1* and *klf-3* led to a mild suppression of longevity in *isp-1;ctb-1* mutants, again, likely due to dilution of both RNAi (Supplementary Fig. 5d).

To address a potential role of KLF-1 in the regulation of autophagy we analysed at the expression of genes encoding proteins involved in the autophagosome formation and did not observe any differences due to mitochondrial dysfunction or depletion of KLF-1 (Supplementary Fig. 5e). Furthermore, the number of GFP-positive puncta in the hypodermal seam cells, commonly used as a marker for autophagosomes, did not change in the absence of *klf-1, klf-3* or both factors in the same time (Supplementary Fig. 5f). Remarkably, we detected higher autophagy flux (cleavage of GFP::LGG-1 inside autophagosome) in *isp-1;ctb-1* mutants only in conditions where KLF-1 was depleted (Supplementary Fig. 5g). The effect was further exaggerated by addition of vitamin C (Supplementary Fig. 5g), suggesting that KLF-1 and its activation by mitochondrial ROS might play a role in suppression and not activation of autophagy[13]. Taken together, these results argue against a major role for the autophagy in *isp-1;ctb-1* mutants and question proposed role for the KLF-1 in this process.

**Cytochrome P450 oxidases are longevity assurance proteins**. The upregulation of CYPs should give advantage to animals upon contact with xenobiotics. To test this, we exposed animals to the known CYP-target vinblastine, a drug that binds tubulin and thus prevents cell proliferation, and assessed the drug-induced developmental delay (Fig. 5a). Remarkably, *isp-1;ctb-1* mutants that normally develop slower than WT animals, upon vinblastine treatment reached further stages in development than controls, an effect fully dependent on KLF-1 (Fig. 5a). Moreover, *isp-1;ctb-1* showed KLF-1-dependent, increased resistance to levamisole-induced paralysis (Fig. 5b). These data suggest that the high expression of *cyps* driven by KLF-1 provides higher resistance to xenobiotics in the *isp-1;ctb-1* that might be essential for longevity. Furthermore, *isp-1;ctb-1* animals showed high resistance to acute, high-dose $H_2O_2$ treatment that was fully dependent on KLF-1 and CYPs presence (Fig. 5c).

The knockdown of two highly upregulated *cyps* (*cyp-13a11* and *cyp-25a1*) strongly suppressed the *isp-1;ctb-1* longevity, indicating an important longevity assurance function of these proteins (Supplementary Fig. 6a). Remarkably, the same effect was observed when *cyps* are depleted only during adulthood, mirroring the results obtained for KLF-1 and further strengthening their interdependence in the longevity pathway mediated by mitochondrial dysfunction (Fig. 5d).

To further evaluate transcriptional control of the xenobiotic response in mitomutants, we used reporter strains in which the *gfp* expression was driven either by *cyp-25a2* or *cyp-34a8* promoter (Fig. 5e and Supplementary Fig. 6b). Predictably, the *gfp* levels progressively increased in the *isp-1;ctb-1* through adulthood in a KLF-1-dependent manner. In contrast, a *skn-1* knockdown, a homologue of mammalian NRF2 (Nuclear factor (erythroid-derived) 2)-like 2), a transcription regulator of phase II detoxification response[22,23], further increased the expression of both reporters (Fig. 5e and Supplementary Fig. 6b). This supports the idea that in the absence of SKN-1, which regulates phase II and the overlapping antioxidant response, endogenous toxic compounds that are produced and not excreted, would further exaggerate phase I response.

Inversely, KLF-1 exclusively regulates a phase I, and not phase II response as shown by the analysis of the phase II/antioxidant gene *gst-4* expression upon treatment with acrylamide, a compound that notoriously activates a phase II xenobiotic response[24]. We show that the acrylamide treatment of WT worms strongly activates *gst-4* expression independently of KLF-1, and this upregulation is fully suppressed by SKN-1 depletion (Fig. 5f and Supplementary Fig 6c, d). The *gst-4* expression was not upregulated in *isp-1;ctb-1* (Fig. 5f and Supplementary Fig 6c, d) and SKN-1 depletion had only a mild effect on *isp-1;ctb-1* longevity (Supplementary Fig. 6e). Nevertheless, the expression of phase II genes (*gst-4* and F56A4.4, a *gst-10* homologue) in *isp-1; ctb-1* was strongly dependent on both KLF-1 and CYPs presence (Fig. 5g and Supplementary Fig. 6f). This is in agreement with previous results showing that phase I detoxification reactions, catalysed by CYPs, result in the addition of a reactive group onto the toxic compound that is required for the stimulation of phase II metabolism[3].

Together our data support the view that KLF-1 directly regulates phase I and not phase II response genes, which instead depend on SKN-1. However, they also signify the necessity of phase I enzymes for the phase II activation and provide an explanation of how KLF-1 mediated activation of *cyps* regulates the level of oxidative damage and antioxidant response.

**ROS pulse in development is essential for KLF-1 activation**. To further understand the involvement of KLF-1 in the mitohormetic response and its specific activation, we treated animals with a low amount of paraquat that has been shown to induce longevity in WT animals through the same mechanisms described for mitomutants[15,25]. The observed longevity phenotype was fully suppressed in a *klf-1(tm1110)* deletion mutant (Fig. 6a). Remarkably, we show that paraquat treatment administered exclusively during development is also able to induce longevity that is suppressed by KLF-1 depletion during adulthood, but not during development (Fig. 6b–d), thus mirroring the results obtained from the *isp-1;ctb-1* mutants (Fig. 1b, c).

We next measured mitochondrial ROS production at different time points and detected higher levels of $H_2O_2$ production in *isp-1;ctb-1* at the fourth larval stage (L4), comparing with WT (Fig. 6e). Interestingly, while the mitochondrial ROS production in WT worms more than tripled on the D1, we observed only a mild increase in *isp-1;ctb-1* mutants, in agreement with the mitohormetic theory (Fig. 6e).

We have previously shown that mitochondrial biogenesis for all somatic tissues occurs predominantly between late L3 and late L4 stage and finishes around the time of transitioning into adulthood[26]. Consistently, increased mitochondrial mass was observed on the D1 in both WT and mutant worms, with higher upregulation in the *isp-1;ctb-1* that was not dependent on KLF-1 (Fig. 6f and Supplementary Fig. 7a). Therefore, despite mitochondrial dysfunction and higher ROS levels during larval development, *isp-1;ctb-1* mutants generated less ROS per mitochondria once they reached adulthood (Fig. 6g). This was maintained also later in adulthood and was clearly KLF-1 dependent (Fig. 6h). Remarkably, KLF-1 depletion at D5 caused much higher ROS burst in WT animals than in *isp-1;ctb-1*, again supporting the mitohormetic theory (Fig. 6h).

We next show that the majority of the KLF-1, under normal conditions, is located in the cytoplasm, with nuclear localization observed only in some intestinal cells (Fig. 7a, b). In the *isp-1;ctb-1* mutant, or upon mild oxidative stress induced by treatment with either paraquat or antimycin A, KLF-1 translocates to the nucleus (Fig. 7a, b). As we observed a significant increase in ROS production in the *isp-1;ctb-1* late developmental stages (L4), we questioned whether redox signalling might be involved in the activation of KLF-1. Indeed, treatment with antioxidants, e.g. N-acetyl cysteine (NAC) and vitamin C, completely abolished KLF-1

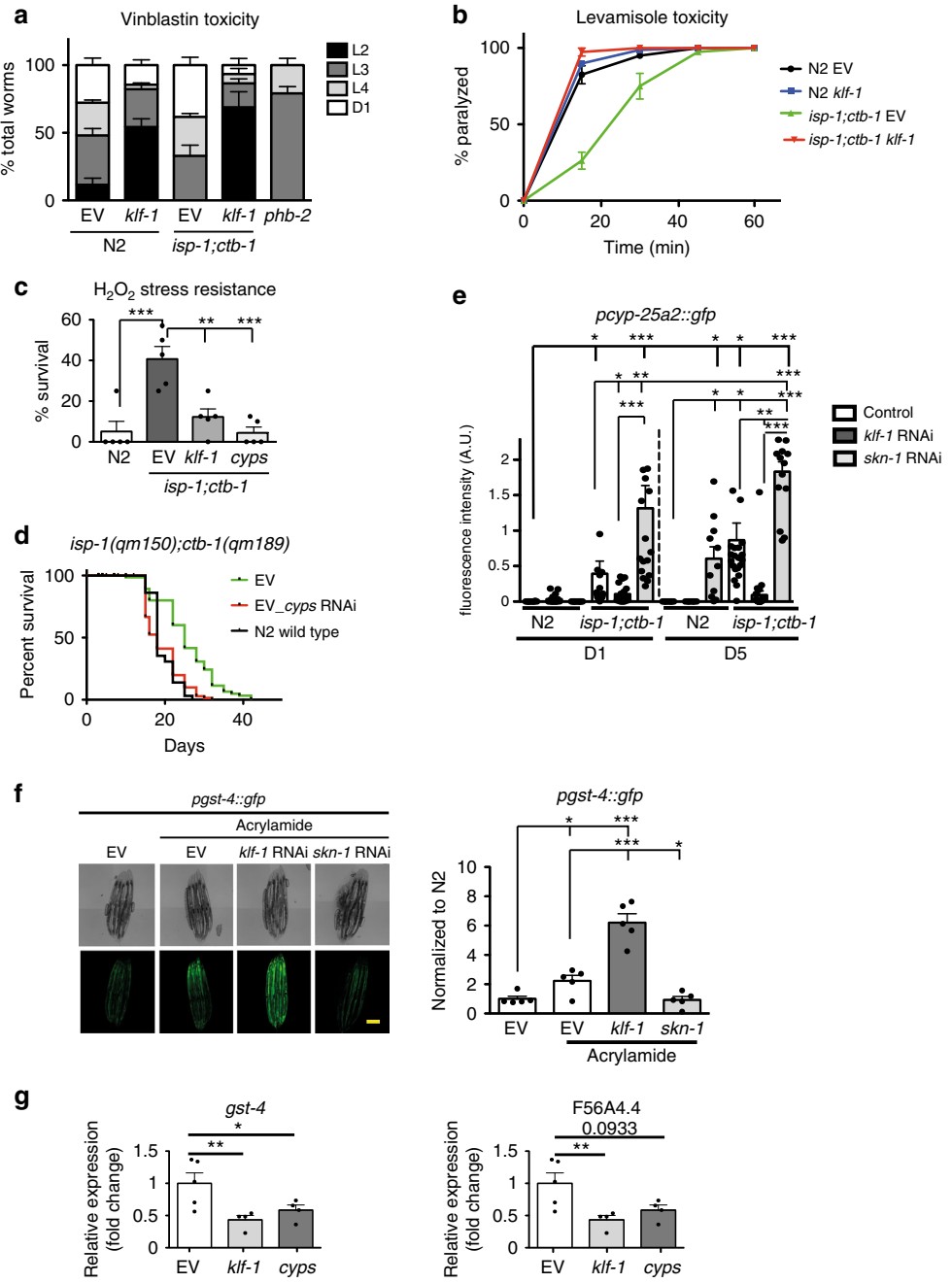

**Fig. 5** KLF-1 regulates phase I, but not phase II detoxification pathway. **a** L1 larvae of N2 wild type, *klf-1(tm1110)*, *isp-1(qm150);ctb-1(qm189)*, *isp-1(qm150);ctb-1(qm189);klf-1(tm1110)* and *phb-2(ad2154)* were grown in liquid with and without 100 µM vinblastin. When all animals without treatment reached adulthood, the animals in the wells containing vinblastin were assayed for developmental stages. *n* = 5 wells per condition per experiment. Data presented are average of three separate experiments. **b** N2 wild-type and *isp-1(qm150);ctb-1(qm189)* animals were transferred at the fourth day of adulthood on plates containing 1 mM levamisole and assayed every 15 min for movement. Worms that failed to move upon gentle touch with silver wire were considered paralysed. *n* = 4 plates with 25 worms for each condition. **c** N2 wild type and *isp-1(qm150);ctb-1(qm189)* animals on control (EV), *klf-1* and combined *cyp-25a1* and *cyp-13a11* RNAi (*cyps*) animals were treated with 20 mM $H_2O_2$ at the first (D1) day of adulthood and assayed for survival 4 h later. **$p < 0.01$, ***$p < 0.001$, one-way ANOVA with Tukey post hoc test. *n* = 100 animals per condition. **d** Survival curve of N2 wild type and *isp-1(qm150);ctb-1(qm189)* animals on control (EV) plates and combined *cyp-25a1* and *cyp-13a11* RNAi (*cyps*). Animals were exposed to RNAi from L4 larval stage. **e** Quantification of *gfp* expression under *cyp-25a2* promoter in N2 wild type and *isp-1(qm150);ctb-1(qm189)* mutant background upon *klf-1* and *skn-1* RNAi at D1 and the fifth (D5) day of adulthood. Data are presented as mean ± SEM. *$p < 0.05$, **$p < 0.01$, ***$p < 0.001$, one-way ANOVA with Tukey post hoc test. *n* = 10 animals per condition. **f** Left panel shows representative confocal images of *gfp* expressed under *gst-4* promoter in wild type animals. Animals were treated with acrylamide upon *klf-1* and *skn-1* RNAi at D1. Scale bar 100 µm. Right panel is the quantification of *n* = 5 animals per condition. Data are presented as mean ± SEM. *$p < 0.05$, ***$p < 0.001$, one-way ANOVA with Tukey post hoc test. **g** SKN-1 transcriptional targets *gst-4* (left) and F56A4.4 (right) expression levels were quantified by qPCR in *isp-1(qm150);ctb-1(qm189)* animals upon *klf-1* RNAi or combined *cyp-25a1* and *cyp-13a11* RNAi (*cyps*). Data are presented as mean ± SEM. *$p < 0.05$, **$p < 0.01$, Student's T-test. *n* = 4 independent samples per condition

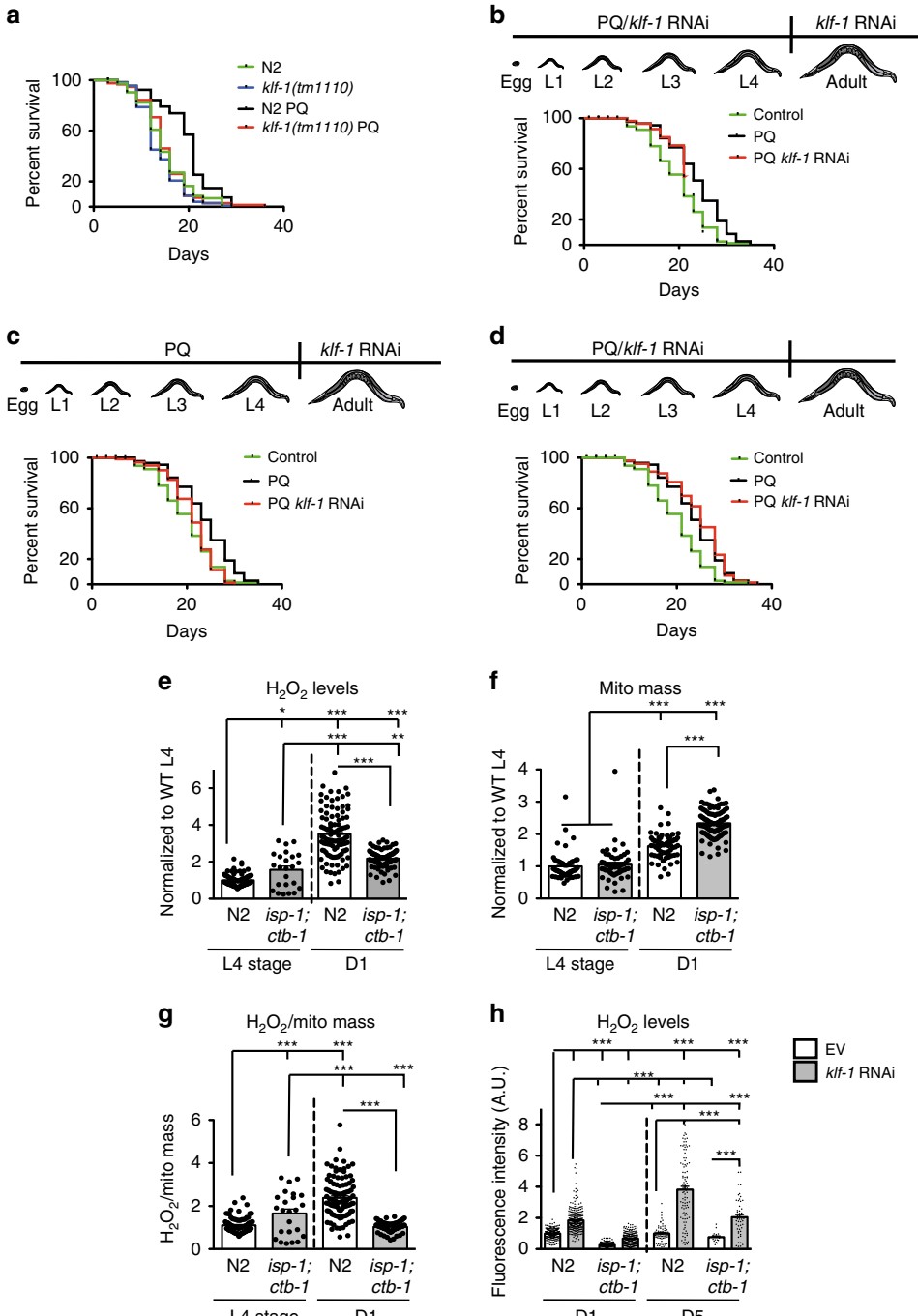

**Fig. 6** KLF-1 mediates a mitohormetic response in *isp-1(qm150);ctb-1(qm189)* mutants. **a** Lifespan curves of N2 wild-type and *klf-1(tm1110)* animals grown on 0.1 mM paraquat (PQ) their whole life. **b**–**d** Worms were grown on 0.1 mM paraquat (PQ) until L4 larval stage and then transferred to paraquat-free plates. Worms were exposed to *klf-1* RNAi either whole life (**b**), in adulthood (**c**) or during development (**d**). **e**–**g** N2 wild-type and *isp-1(qm150);ctb-1(qm189)* animals were stained with Mitotracker Red CM-H$_2$XRos (**e**) or Deep Red (**f**) to assay mitochondrial ROS levels and mitochondrial (mito) mass, respectively. Animals were assayed at larval L4 stage or D1 and the fluorescence was measured via Biosorter. A ratio between Mitotracker Red CM-H$_2$XRos and Deep Red fluorescence measurements is shown in (**g**). **h** Mitotracker Red CM-H$_2$XRos staining of N2 wild type and *isp-1(qm150);ctb-1(qm189)* animals grown on control (EV) or *klf-1* RNAi plates and assayed at the D1 or D5 of adulthood. Data are presented as mean ± SEM. ***$p < 0.001$, one-way ANOVA with Tukey post hoc test. $n = 60$ animals per condition

translocation to the nucleus induced by antimycin A treatment or *isp-1;ctb-1* mutations, suggesting that KLF-1 activation is indeed regulated by redox signalling (Fig. 7c). Remarkably, in WT animals KLF-1 is predominantly present in the nucleus at L4 stage, to be quickly excluded upon transition to early

adulthood (Fig. 7d). In the *isp-1;ctb-1* animals we observed persistent KLF-1 localization at D1 that is largely reduced by the treatment with vitamin C, providing further evidence that oxidative stress signalling plays important role in this process (Fig. 7d). A depletion of SKN-1 also induced KLF-1 translocation

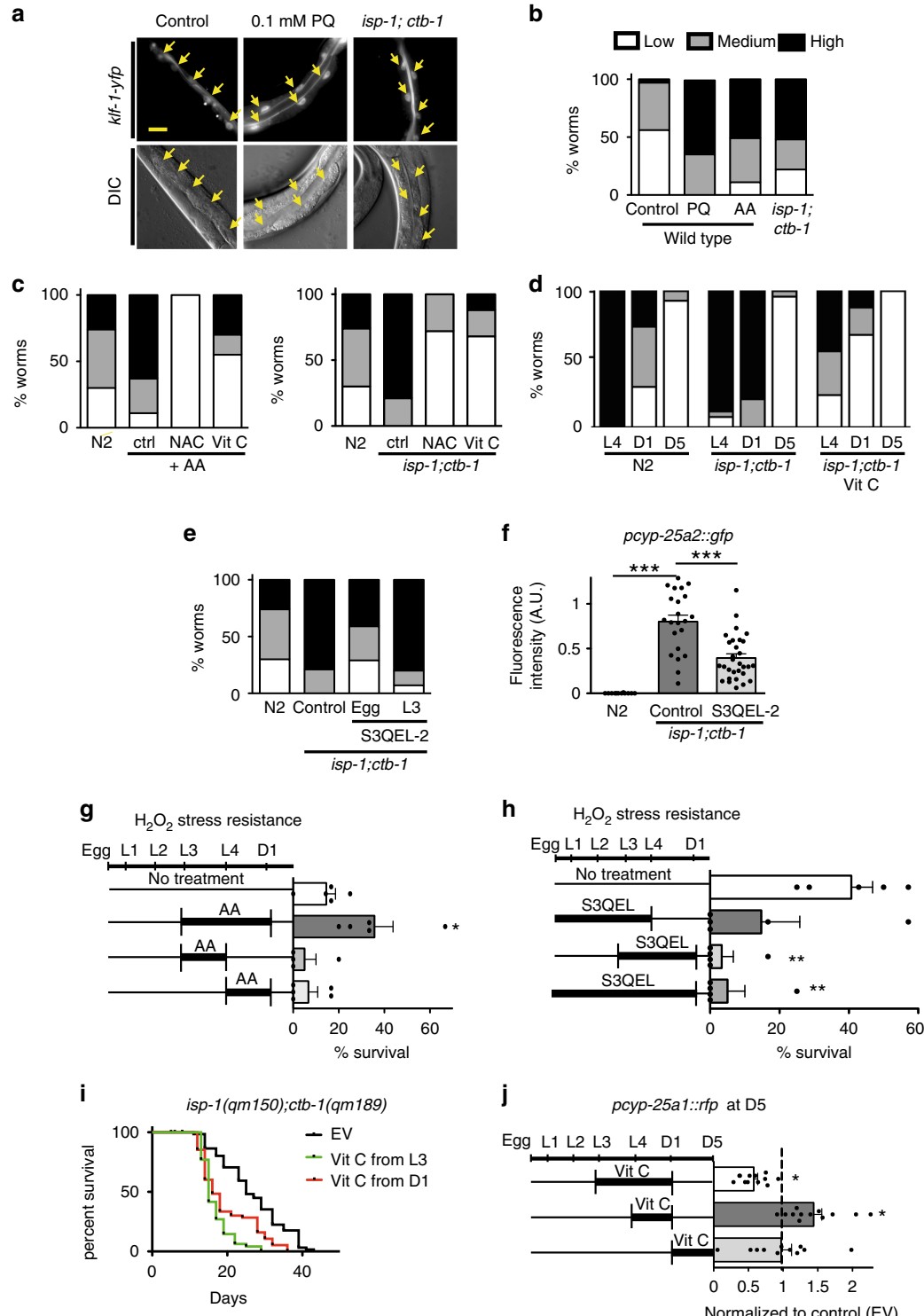

to the nucleus in control animals, but not in *isp-1;ctb-1* mutants (Supplementary Fig. 7b). As SKN-1 is a major regulator of antioxidant-enzymes[22], this effect in WT is likely a consequence of higher ROS levels, while this response is epistatic in *isp-1;ctb-1* mutants.

Mutations in *isp-1* and *ctb-1* lead to decreased electron flow through respiratory complex III and increased ROS production[15]. To show that the initial redox signal needed for KLF-1 activation originates from mitochondrial ROS, we treated the *isp-1;ctb-1* animals with S3QEL-2, a compound that specifically blocks ROS production from complex III, although not complex I[27]. Indeed,

this treatment largely prevented the KLF-1 translocation to the nucleus and therefore the CYPs activation (Fig. 7e, f).

Remarkably, we could map the initial ROS signal needed for the translocation/activation of KLF-1 to the L3–D1 transition, which directly coincides with the peak in somatic mitochondrial biogenesis, as we previously described[26]. Hence, the KLF-1 translocation could be prevented by the administration of S3QEL-2 from the egg, but not if the treatment started at late L3 stage (Fig. 7e). The essential role of the L3 to young adulthood period for the activation of protective stress responses in *isp-1;ctb-1* mutant is further supported by our finding that treatment with

**Fig. 7** KLF-1 activation is mediated by redox signalling. **a** Representative images of KLF-1-YFP, expressed under gut specific *vha-6* promoter. Arrows indicate gut nuclei. Scale bar is 200 µm. **b**–**e** Animals were assayed based on KLF-1 nuclear localization as follows: "low" as less than 2 nuclei, "medium" 3–10 nuclei, and "high" where all nuclei were stained. Animals were imaged at D1, unless otherwise stated. $n = 20$ animals per condition. **b** WT animals were grown on control or plates containing 0.1 mM paraquat (PQ), or 1 µM Antimycin A (AA). **c** N2 animals were grown on control or plates containing 1 µM antymicin A or antymicin A in combination with 10 mM NAC or 10 mM vitamin C (left). Right, *isp-1(qm150);ctb-1(qm189)* were grown on 10 mM NAC or 10 mM vitamin C. As control, KLF-1-YFP nuclear localization in WT was used. **d** KLF-1-YFP expressing animals in N2 or *isp-1(qm150);ctb-1(qm189)* were assayed at L4 stage, D1 or D5, with or without vitamin C treatment. **e** *isp-1(qm150);ctb-1(qm189)* were grown on 100 µM S3QEL-2 either from egg stage or L3 stage and assayed for KLF-1 nuclear localization at D1. **f** Quantification of *rfp* expression under the *cyp-25a2* promoter in N2 and *isp-1(qm150);ctb-1(qm189)* upon treatment with 100 µM S3QEL-2 during whole development at the second day of adulthood. Data are presented as mean ± SEM. ***$p < 0.001$, one-way ANOVA with Tukey post hoc test. $n = 10$ animals per condition. **g** N2 animals were treated with 20 mM $H_2O_2$ at D1 and survival was assayed 4 h later. Animals were treated with 2 µM antymicin at designated developmental stages. Data are presented as mean ± SEM. *$p < 0.05$, Student's T-test. $n = 100$ animals per condition. **h** *isp-1(qm150);ctb-1(qm189)* were treated with S3QEL-2 at designated developmental stages and the animals were assayed for $H_2O_2$ resistance at D1. Data are presented as mean ± SEM. **$p < 0.01$, Student's T-test. $n = 100$ animals per condition. **i** Lifespan curve of *isp-1(qm150);ctb-1(qm189)* grown on control (EV) or treated with 10 mM vitamin C from either L3 stage or D1. **j** Fluorescence quantification from *pcyp-25a2::rfp* reporter strain. *isp-1(qm150); ctb-1(qm189)* were assayed at D5, upon the designated treatments with 10 mM vitamin C. Data are normalized to control animals (presented as dashed line). *$p < 0.05$, Student's T-test. $n = 10$ animals per condition

antymicin A, a potent inducer of ROS production at complex III, during this specific period, paradoxically, provides strong resistance to oxidative stress (Fig. 7g). Conversely, treatment with S3QEL-2 during the same period largely decreased $H_2O_2$ resistance in *isp-1;ctb-1* mutants (Fig. 7h). Consistently, block of ROS production from L3 fully suppresses the longevity in *isp-1; ctb-1*, unlike suppression from D1 that had only a partial effect (Fig. 7i). This is further in agreement with finding that vitamin C treatment also decreases expression of *cyp-25a2* reporter when administered during the L3–D1 period (Fig. 7j).

Therefore, our data strongly imply that ROS directly or indirectly modifies KLF-1 resulting in a subsequent nuclear translocation and activation of target genes. In line with this, we show that KLF-1 overexpression leads to only a moderate increase in lifespan, with little effect on analysed target genes (Supplementary Fig 7c, d). However, the increase was fully dependent on the *cyps* expression (Supplementary Fig. 7e). We could further show that the *klf-1* overexpressors are more resistant to treatment with high $H_2O_2$ levels (Supplementary Fig. 7f).

Collectively, these results suggest that longevity assurance adaptations that are regulated by KLF-1, although initiated by ROS, activate more elaborate stress responses rather than directly combating oxidative stress. This complex adaptation then contributes to mitohormesis that renders animals more resistant to toxic endogenous and xenobiotic compounds leading to an increased longevity.

## Discussion

The upregulation of genes involved in xenobiotic detoxification has been increasingly recognized as a transcriptional signature of many long-lived mutants of different species[1,7,28,29]. Mice harbouring various life-promoting mutations in growth hormone pathway, including Ames dwarf mice, Snell dwarf mice, GHRKO mice and Little mice, all show a transcriptomic signature of elevated xenobiotic detoxification genes[5,30]. Different pharmacological and dietary interventions that promote longevity in mice also increase expression of DME encoding genes, including mice subjected to dietary restriction, reduced access to the mother during the breastfeeding period ('crowded litter'), or treated with rapamycin[5,31]. In agreement, many of them also showed increased resistance to different xenobiotics, including hepatotoxins[32–34]. These findings suggest that increased metabolism of endo- and xenobiotics might be a downstream mechanism mediating the effects of multiple life-prolonging interventions. Here, we identify KLF-1 as the first transcription factor regulating this response that is essential for a longevity mediated by reduced

mitochondrial function. The longevity inducing signal, arising from mitochondrial dysfunction, coincides with a time when majority of somatic mitochondrial biogenesis occurs, between the L3 and D1 stages in worms[26,35]. We show that the abrupt increase in the number of dysfunctional mitochondria in mutant worms produces a signalling pulse of mtROS, which facilitates nuclear translocation, and therefore, activation of the KLF-1-mediated response. Remarkably, this is also the time when the aerobic respiration peaks[36] and it was previously reported that downregulation of mitochondrial function only during this period is sufficient to prolong the lifespan[37]. Therefore, we propose that this specific period has been set as an important checkpoint for mitochondrial function that is carefully monitored as vital for the adult worm. A safety mechanism in the form of ROS-regulated stress response mediated by KLF-1 was also put in place to ensure a damage control, upregulation of detoxification response to safeguard proper functioning during the adulthood. We further show that the KLF-1 activation has to be timed to D1, in order to support reproduction and provoke the life-extending mitohormetic response observed in mitomutants.

KLFs moderate many different fundamental processes in the cell[38]. Intriguingly, many of the mammalian 17 KLF homologues are involved in different kind of stress responses or are essential for switching developmental to adult gene programmes, as in the case of beta-globin genes[38]. Our results argue that both of these functions are preserved in KLF-1. The closest homologues (KLF2, KLF4 and KLF5) of nematode KLF-1 are also highly responsive to oxidative stress and seem to have complementary, very precisely timed functions in mammalian cardiomyocytes and vascular endothelium[13,39]. The dynamic regulation of expression of multiple KLF family members suggests that they are actively involved in regulating phenotypic responses to extracellular stimuli[39]. Furthermore, some members of mammalian KLF family are shown to be important for mitochondrial function in different tissues, including KLF6, that is essential for mitochondrial gene expression and morphology in kidney or KLF4 and KLF15 that seem to regulate mitochondrial biogenesis and function in heart[40–42].

Our study also identified CYPs as direct effectors of the KLF-1 mediated response that promotes longevity. In mammals, the majority of *cyp* expression is governed by a very complex regulation process that includes four nuclear transcription factors, two different cofactors, and an elaborate signalling cascade[43]. Intriguingly, some of mammalian KLFs were also shown to bind the basic transcription element (BTE) in different *cyp* promoters, and therefore, potentially regulate their expression[44,45]. In fact, some KLFs were initially identified as BTEB (BTE-binding)

proteins and only later renamed (BTEB/KLF9, BTEB2/KLF5, BTEB3/KLF13, BTEB4/KLF16)[46–48]. Recently, mammalian KLF6 was identified as a novel DNA binding partner of the aryl hydrocarbon receptor (AhR), a bona fide transcription factor that activates expression of many *cyps* and *ugts* through binding to xenobiotic response elements (XRE)[49], while KLF9 was shown to regulate hepatic CYP2D6 expression during pregnancy[50]. However, very little is known about the physiological relevance of this regulation. Similarly, potential effect of KLFs on mammalian ageing has not been explored although they are involved in the regulation of many molecular and cellular hallmarks of aging as well as age-associated diseases[51]. We also demonstrated that both KLF-1 and CYPs are needed for the activation of phase II genes, mainly enzymes of glutathione metabolism[22]. However, KLF-1 does not directly regulate phase II genes and evidently SKN-1 does not regulate the expression of most *cyp* genes, as shown here and by others[52].

Xenobiotic/detoxification pathways are increased in *isp-1;ctb-1*, resulting in increased resistance toward different drugs, in agreement with reports on other mitomutants that exhibit resistance to multiple drugs[53]. Even a simultaneous knockdown of both mitochondrial SODs (SOD-2 and SOD-3) boosted the resistance to multiple drugs, suggesting that an increase in mitochondrial ROS is needed for this adaptation[53]. Previous studies have detected increased expression of genes encoding proteins involved in xenobiotic detoxification, including *cyp*, *ugt* and *gst* genes, in long-lived mitochondrial mutants[6,7,28]. Since many bacterial species from *C. elegans* habitats target mitochondria, the evolutionary advantage of coupling detection of mitochondrial dysfunction to antibacterial gene expression is clear[7]. However, other longevity promoting pathways, such as decreased insulin/IGF1 signalling or dietary restriction, also activate the expression of DMEs in a wide array of organisms ranging from worms[29,54] to mammals[5]. Therefore, it is possible that coupling of detoxification pathways to longevity has arisen across different mechanisms that are regulated by similar or even overlapping pathways. Suitably, KLF-1 seems to be essential for the longevity in the mitomutants, upon dietary restriction and suppression of insulin signalling pathway, although different set of genes have been proposed to be essential for this effect[12,13].

In summary, our results provide clear evidence that KLF-1 regulates longevity assurance pathway through its role in activation of genes involved in xenobiotic detoxification. This pathway seems to interconnect with many mechanisms of lifespan extension, including mitochondrial dysfunction, caloric restriction and insulin/IGF-1 signalling. We show that the initial signal in mitochondrial mutants is precisely timed with a mild increase in mtROS production. Importantly, our results also provide an answer to a long-standing puzzle in the field: Why have studies of antioxidant genes failed to reveal a major causal role in the regulation of lifespan, despite the fact that most long-lived animals show higher resistance to oxidative stress? Our study now suggests that increased resistance to oxidative stress in long-lived animals stems from the activation of a complex xenobiotic detoxification pathway shown to be upregulated in by all major longevity models. In view of this study, future research efforts should focus on establishing whether our findings extend to mammals as a prelude to developing novel interventions and new therapeutic strategies to combat aging in general and age-associated diseases in particular.

## Methods

**Strains**. Strains were cultured on OP50 *E. coli*-seeded NGM plates, according to standard protocols unless otherwise stated[55]. The following strains were used in this study: Bristol N2, MQ887 *isp-1(qm150)*, MQ989 *isp-1(qm150);ctb-1(qm189)*, CB1370 *daf-2(e1370)*, SJ4100 N2;*zcIs13* [*hsp-6::gfp*], ATR1010 *isp-1(qm150);ctb-1*

*(qm189);zcIs13, klf-1(tm1110)*, GR1373 *eri-1(mg366)*, ATR2640 *isp-1(qm150);ctb-1 (qm189);eri-1(mg366)*, TK22 *mev-1(kn1)*, CW152 *gas-1(fc21)*, CL2166 N2;*dvIs19 [pgst-4::gfp-nls]*, ATR1040 *isp-1(qm150);ctb-1(qm189);dvIs19*, SD1444 *gaIs237 [cyp-25a2p::his-24::mCherry;unc-119(+)]*, ATR4029 *isp-1(qm150); ctb-1(qm189); gaIs237*, CF1553 *muIs84 [(pAD76)sod-3p::gfp;rol-6(su1006)]*, ATR4051 *isp-1 (qm150);ctb-1(qm189);muIs84* ATR4052 *daf-2(e1370);muIs84*, DA2154 *phb-2 (ad2154)*, DA2123 N2;*adIs2122 [plgg-1::gfp-lgg-1]* and ATR1043 *isp-1(qm150);ctb-1 (qm189);adIs2122*.

ATR1022 N2; *atEx100 [pklf-1::gfp;rol-6(su1006)]* and ATR4030 N2; *atEx4030 [pcyp-34a8::gfp;prab-3::mCherry]*, were generated by injecting the *pklf-1::gfp* (50 ng/ µl) and *pcyp-34a8::gfp* (50 ng/µl), respectively, with pRF4 (50 ng/µl) plasmids into the N2 wild-type animals. ATR1022 and ATR4030 were crossed into *isp-1(qm150); ctb-1(qm189)* to create ATR4050 *isp-1(qm150);ctb-1(qm189);atEx100* and ATR4026 *isp-1(qm150);ctb-1(qm189);atEx4030*. ATR4086 N2;*atEx4086 [pvha-6::klf-1-yfp; prab-3::mCherry;rol-6(su1006)]* was created by using *pvha-6::klf-1-yfp* (40 ng/µl), pGH8 (20 ng/µl) and pRF4 (40 ng/µl) plasmids. Plasmid mixtures were then injected using standard procedures[56]. ATR4086 was subsequently integrated and outcrossed five times (ATR4081 strain) before being crossed into *isp-1(qm150);ctb-1(qm189)* to create ATR4082 strain.

**RNAi treatment**. RNAi knockdown was performed as described previously[57]. All genes for RNAi were obtained from the Ahringer RNAi library[57] and confirmed by sequencing. As a control, empty L4440 vector was used. All clones were transformed into the *E. coli* HT115 (DE3) strain. The overnight culture grown in Luria broth media was grown to $OD_{595} = 0.5$, and then IPTG was added to a concentration of 1 mM. The bacteria were then induced for 3 h at 37 °C, shaking and seeded on NGM plates containing 100 µg/ml ampicillin and 1 mM IPTG. Worms were treated with RNAi from hatching and phenotype was observed as indicated. In order to obtain 5-day-old worms, beginning from the first day of adulthood, worms were washed every day with M9 and allowed to settle by gravity, in order to remove eggs and lighter larvae.

**RNAi Screen**. For the *isp-1(qm150);ctb-1(qm189)* longevity suppression screen, RNAi clones from chromosome III of the Ahringer library[8,57] were inoculated in 384-deep well plates and grown overnight. Bacteria were then seeded on 24-well NGM plates containing 100 µg/ml ampicillin and 2 mM IPTG. Each clone was seeded on four wells and induced overnight at room temperature. Afterwards, 15–20 eggs were placed in each well, and the worms were grown at 25 °C. To avoid separating the parental generation from the progeny, the *isp-1(qm150);ctb-1 (qm189)* strain was crossed into the *eri-1(mg366)* mutant background, since this mutation leads to embryonic lethality at 25 °C. As a control, *eri-1(mg366)* and *isp-1 (qm150);ctb-1(qm189);eri-1(mg366)* strains were grown on a separate 24-well plate, on L4440. Nine to ten days after reaching adulthood, 50% of the *eri-1(mg366)* population was scored as dead. On that day, survival of *isp-1(qm150);ctb-1(qm189); eri-1(mg366)* was assayed and the RNAi clones that gave a similar phenotype as the *eri-1(mg366)* single mutant, were identified. Among all the candidates identified, transcription factors were retested using *eri-1(mg366)* and *isp-1(qm150);ctb-1 (qm189);eri-1(mg366)* strains, at 25 °C.

**Lifespan analysis**. For lifespan analysis, worms were grown at 20 °C, unless otherwise stated. For the experiments with paraquat, the drug was added to the plates at a concentration of 0.1 mM and worms were exposed to the drug either their whole life or from hatching until L4 larval stage. First day of adulthood was defined as day 1 of lifespan. Unless otherwise stated, 25 worms were transferred to each plate at day 0, for a total of 100–120 worms per experiment. Worms were examined every second day by prodding with a silver wire. The worms that escaped the plate, or died due to internal hatching or protrusions, were censored.

For the "switch" experiments, worms were grown either on empty L4440, *klf-1*, *cyc-1*, *atp-5* or *cco-1* RNAi until the L4 larval stage or the third or fifth day of adulthood, as stated. Worms were then transferred to either L4440 or *klf-1* RNAi plates. Compilation of lifespan assays is listed in Supplementary Data 4.

**Oxidative stress sensitivity assay**. In order to avoid bagging or *egl* phenotypes in worms, upon exposure to high oxidative stress, experiments were performed on strains with an *eri-1(mg366)* background at 25 °C, since this mutation leads to embryonic lethality at this temperature. At the L4 larval stage, the first and the fifth day of adulthood, worms were transferred onto NGM plates containing 16 mM paraquat. Survival of worms was checked every 8–12 h.

**$H_2O_2$ resistance assay**. One-day-old adult worms were picked into 96-well plates filled with M9 buffer. $H_2O_2$ was then added to final concentration of 20 mM. Animals were scored for survival every hour. Presented are the data 4 h after treatment. For each condition, 12 wells of 8 worms each were assayed.

**Post embryonic development**. Worms were exposed to RNAi from hatching and scored every 8 h for vulva formation. The total time from hatching until the vulva formation was taken as developmental time.

**Brood size**. Single worms were transferred on individual plates at the L4 larval stage and allowed to lay eggs. The worms were transferred to a fresh plate every day until they stopped laying eggs. The total amount of hatched progeny was scored and plotted as total brood size.

**Movement**. Worms were grown on indicated RNAi plates. On the first and the fifth day of adulthood, worms were transferred to non-seeded NGM plates, and allowed to settle for an hour. Afterwards, movement was scored as the number of full sinusoidal curves, body bends that a worm made moving forward or backwards, over a period of 3 min. The data were presented as the number of body bends per minute.

**KLF-1 localization assay**. To assess KLF-1 intracellular localization, ATR4081 and ATR4082 strains were used. For antioxidant treatments, animals were grown from eggs or as otherwise stated on 10 mM NAC (Sigma Aldrich), 10 mM vitamin C (Sigma Aldrich) and 100 μM S3QEL-2 (Caymann Chemicals). The chemicals were added directly to the NGM plates prior to seeding with HT115 bacterial strain. For antimycin A treatment, antimycin A was added directly to the NGM medium to final concentration of 1 μM. These plates were than seeded with HT115 bacteria and allowed to dry from couple of days. Animals were grown until L4 larval stage and then transferred to the fresh NGM plates containing antimycin A. All animals were imaged as young adults unless otherwise stated. Activation was quantified as "low" when less than 2 nuclei per animal were stained, "medium" when 2–6 nuclei were stained, and "high" when more than 6 nuclei were observed. Each experiment was repeated at least three times on different days.

**ROS measurements**. For ROS measurements and mitochondrial staining Mitotracker Red CM-$H_2$XRos and Mitotracker Deep Red were used, respectively. Animals were grown until indicated stages. Day before the experiments, NGM plates were seeded with heat-inactivated bacteria and let dry overnight on room temperature. On the day of the experiments, 200 μl of 10 μM dye solutions were added on top of the bacteria and let dry. Animals were then added to the plates and incubated for 1 h in the dark. Afterwards, animals were washed few times with M9 buffer and transferred to fresh plates without the dye for 2 h. Subsequently, worms were washed and analysed with Biosorter Instrument (Union Biometrica).

**Xenobiotic resistance assays**. For vinblastin treatment, ~20–30 synchronized L1 larvae were incubated in S-Complete with 6 mg/ml HT115 bacteria. Wells were divided into controls without vinblastin and with 100 μM vinblastin (Sigma Aldrich). Worms were checked every day. When animals in the control wells reached adulthood, the animals treated with vinblastin were scored for developmental stages. At least five wells were used per condition and experiment was repeated three times.

For levamisole assay, levamisole was added directly to the NGM plates to final concentration of 1 mM. Animals were transferred to these plates on the second day of adulthood and scored every 15 min for paralysis by prodding gently with the silver wire. If no movement was observed, animals were scored as paralysed. A minimum of five plates were used per condition with 25 worms each.

**Microarray analysis**. For the microarray experiment, RNA was isolated from worms collected from a 9-cm plate, using the Qiagen RNeasy kit. Conditions and strains used are as follows: N2 and isp-1(qm150);ctb-1(qm189) grown on L4440 plates, isp-1(qm150);ctb-1(qm189) grown on klf-1 RNAi plates from hatching to the L4 larval stage and then switched to L4440 plates and isp-1(qm150);ctb-1(qm189) grown on L4440 from hatching to the L4 larval stage and then switched to klf-1 RNAi. All worms were collected on the fifth day of adulthood. For each condition, three independent isolates were used. Reversely transcribed RNAs were hybridized to Affymetrix EleGene 1.0 ST microarrays, according to the manufacturer's instructions. Affymetrix CEL files were processed with the Affymetrix Power Tools, version 1.15.2 and the Robust Multiarray Average normalization algorithm[58]. Analysis of differentially regulated genes was carried out using the R language for statistical computing, version 3.0.2. For functional annotation and GO term enrichment analysis, the Database for Annotation, Visualization and Integrated Discovery (DAVID), version 6.7 was used[18]. For the data presented in Supplementary Tables 1 and 2 and Supplementary Data 1, a cutoff of p < 0.05 and fold change >1.2 was used. For the data in the Fig. 4, a cutoff of p < 0.01 and fold change > 1.5 was used. Data availability is stated in Table 1.

**ChiP-Seq assay**. ChiP-Seq assay was performed as described previously[59] using ATR4081 and ATR4082 strains at the first day of adulthood. Crosslinking was performed in 1% formaldehyde for 20 min at room temperature. After lysis and sonication to obtain 300–1000 bp fragments, protein concentration was measured and proteins were normalized to 2 mg in 500 μl volume. Fifty microlitres of the volume was kept as input and the rest of the lysate was incubated with GFP-Trap (Chromotek) for 2 h at 4 °C. After washing, the crosslinks were reversed by incubation at 65 °C overnight. DNA was then isolated using phenol–chloroform–isoamylalcohol extraction and ethanol precipitation protocol.

The DNA fragments were sequenced using Hi-Seq Illumina platform. ChIP-Seq data was analysed by using QuickNGS (Next-Generation Sequencing) pipeline. The ChIP-Seq workflow takes advantage of BWA for genomic alignment of the reads. Reads was mapped to the Caenorhabditis_elegans (Ensembl database version 93). Quality check of the sequencing data was performed with FastQC version 0.10.1. For peak calling used MACS2 version 2.0.10. QuickNGS pipeline identifies all the genes which are 2000 bp up- or downstream from the MACS2 peaks. The results comprise lists of significant peaks. Result was uploaded into MySQL database. Alternatively, ChiP-Seq data were analysed with qPCR using primers listed in Supplementary Data 5. CHIP-seq data are available at NCBI Gene Expression Omnibus (GEO) under GSE130035.

**RNA isolation and qPCR**. Worms were collected from a 9 cm plate and total RNA was isolated with Trizol (Invitrogen). DNAse treatment was performed using DNA-free$^{TM}$, DNAse and removal (Ambion, Life Technologies), according to the manufacturer's protocol. RNA was quantified by spectrophotometry and 0.8 μg of total RNA was reversely transcribed using the High-Capacity cDNA Reverse Transcription Kit (Applied Biosystems). For each condition, five independent samples were prepared. qPCR was performed using the Step One Plus Real-Time PCR System (Applied Biosystems), with the following PCR conditions: 3 min at 95 °C, followed by 40 cycles of 5 s at 95 °C and 15 s at 60 °C. Amplified products were detected with SYBR Green (Brilliant III Ultra Fast SYBR Green qPCR Master Mix, Agilent Technologies). Relative quantification was performed against either act-1 or Y45F10D.4. Primers used in this study are presented in Supplementary Data 5. Data were analysed using ΔΔCt analysis. For each condition, five independent replicates were used. Each gene is tested in two independent experiments. Data presented are average of one experiment.

**klf-1 expression under stress conditions**. ATR1022 strain was grown until adulthood on 20 °C degrees on normal NGM plates and then treated as follows: (i) for osmotic stress animals were transferred to NGM plates containing 500 mM NaCl and kept for 20 h, after which the animals were washed off the plates using M9 buffer containing 300 mM NaCl and transferred to normal NGM plates; (ii) for heat stress, animals were incubated on 35 °C for a period of 9 h; (iii) for oxidative stress, young adults were either transferred to 16 mM paraquat plates for 24 h, or worms were grown from eggs until adulthood on 0.1 mM paraquat throughout the development. At the end of treatment, animals were imaged as described bellow.

**Western blotting**. For protein sample preparations, worms were collected from three full 9-cm plates and wash extensively with M9 buffer, in order to remove the bacteria. The pellets were then frozen in liquid nitrogen. Two hundred microlitres of lysis buffer (25 mM Tris-HCl pH 7.4, 0.15 M NaCl, 1 mM EDTA, 1% NP-40, 0.5% SDS, 10 mM DTT and proteinase inhibitor cocktail) was added to the pellet prior to thawing. Three freeze-thaw cycles were performed in liquid nitrogen, followed by sonication. The debris was spun down for 10 min at 16,000 × g at 4 °C and the supernatant was transferred to a fresh tube. Protein concentration was measured with Bradford assay.

Western blotting was performed using antibodies against MnSOD (1:1000, Upstate #06-984), Grp75 (1:1000, Abcam #82591), HSP-60 (1:2000, BD Transduction Laboratories #611562), GFP (1:2000, Kindly provided by Jan Riemer), HSC70 (1:2000, Santa Cruz #sc-7298) and Tubulin (1:1000, Calbiochem #CP06). Oxyblot analysis was performed using the manufacturer's protocol (Millipore) and the data were normalized to Ponceau S staining of the membrane. All uncropped blots are supplied in the Source Data File.

**Microscopy**. Animals were immobilized on 2% agarose pads in 5 mM levamisole buffer and imaged using an AxioImager Z.1 epifluorescence microscope, equipped with a Hammamatsu camera (OrcaR$^2$) and AxioVision software 4.8. Images were analysed using ImageJ (National Institutes of Health), as previously described[60].

**Oxygen consumption**. Oxygen consumption rates were measured using an Oroboros Oxygraph 2k (Oroboros Instruments GmbH). Three hundred animals, on the first or the fifth day of adulthood, were used for each measurement. Each measurement was performed at 20 °C and repeated at least three times. Data were analysed using DatLab4 software (Version 4.3).

**Hepa1-6 cell experiments**. Hepa1-6 cells (ATCC®-CRL-1830™) were cultured in DMEM medium containing 4 mM L-Glutamine, 1 mM sodium pyruvate, 100 units/mL penicillin-streptomycin and 10% foetal bovine serum (FBS). For the gene silencing of Klf4, Klf5 and negative control 1.2 × 10$^5$ Hepa1-6 cells were transfected in a reverse transfection protocol with Lipofectamine RNAiMAX (Thermo Fisher Scientific) and each siRNA (purchased from Eurogentec) at a dose of 1 nM in 12 wells following the manufacturer's instructions. siRNAs used are as follows: Klf4— GGAACUCUCUCACAUGAAG and Klf5—GACCAUGCGUAAC ACAGAU.

After 24-h cultivation at 37 °C in culture medium, antimycin A (purchased from Sigma Aldrich, 5 mM stock solution in 100% Ethanol, diluted to 2.5 μM working solution in $H_2$O) was added to a final concentration of 50 nM and

incubated for additional 24 h. After the incubation, the cells were collected and used for RNA isolation using Trizol Reagent (Thermo Fisher Scientific) according the manufacturer's instructions. After DNase I treatment (New England Biolabs, UK), the total RNA was reverse-transcribed using the High-Capacity cDNA Reverse Transcription Kit (Thermo Fisher Scientific) with random primer. Quantitative polymerase chain reaction (qPCR) was performed with a QuantStudio 12 K Flex Real-Time PCR System (Thermo Fisher Scientific) in 12 μl aliquots of reaction mixtures containing cDNA, appropriate pairs of primers and Brilliant III Ultra-Fast SYBR®Green QPCR Master Mix (Agilent). Expression levels of the genes were calculated by the comparative CT method using HPRT as endogenous housekeeping genes. Primers used are enlisted in Supplementary Data 5.

**Reporting summary**. Further information on research design is available in the Nature Research Reporting Summary linked to this article.

## Data availability

The source data underlying the figures in the paper are provided as a Source Data file. The Microarray data that support the findings of this study are available in on NCBI Gene Expression Omnibus (GEO) with the GEO Accession Number: GSE61771. CHIP-seq data are available on NCBI Gene Expression Omnibus (GEO) with the GEO Accession Number: GSE130035. All other data are available from the authors upon reasonable request.

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

## Acknowledgements

The authors wish to thank the Caenorhabditis Genetics Center (which is funded by National Institutes of Health Office of Research Infrastructure Programs, P40 OD010440), Shohei Mitani (National Bioresource Project, Tokyo Women's Medical University School of Medicine) and the International *C. elegans* Gene Knockout Consortium for strains; CECAD Imaging Facility for technical assistance. The work was supported by grants of the German Research Council through Collaborative Research Centre 1218 (SFB1218 - TP B01) and European Research Council (ERC-StG-2012-310700).

## Author contributions

A.T. and M.H. conceived the project, designed the experiments, analysed the data and wrote the paper. M.H., E.C., L.B., A.K., J.H., S.M., K.S. and V.P. performed the experiments. P.F. analysed the microarray and CHIP-seq data. All authors commented on the paper.

## Additional information

**Competing interests:** The authors declare no competing interests.

