## [Peer Review File · Nature Communications]

Reviewers' comments:

Reviewer #1 (Remarks to the Author):

The authors have performed a chromosome III-wide screen for genes potentially involved into the longevity phenotype of the double-mutant *isp-1/ctb-1* to identify an evolutionary conserved transcription factor, *klf-1*, to potentially be required for ROS-mediated lifespan extension in such mito-mutants. At the current state, the data are widely associative rather than causal.

As shown in Figure 7, the authors have generated a transgenic overexpressor of *klf-1* under the control of an gut-specific constitutive promoter. The claim that *klf-1* is a transcriptional regulator promoting longevity should be supported by demonstrating that such an overexpressor is long-lived per se.

Additionally and given the potential role of KLF-1 specifically in young-adult worms, the authors should generate a *hsp-16.2* promoter which is gut-specific *and* heatshock-inducible to dissect the issue with developmental timing of expression and/or activity.

The increased levels of ROS in N2s (as shown in this manuscript) should promote longevity in this background. Absence of such effect would fundamentally question the key claim of the manuscript.

Then, such overexpressors should be crossed into *skn-1(zu135)* mutants to prove the claimed independence from NRF2 signaling, to re-test the longevity phenotype.

These findings could potentially extend findings published in Nature Communications in Oct. 2017 (Hsieh et al.) where all nematodal *klf*-genes have been studied in regards to their role in lifespan regulation, but also in a mammalian model.

Apparently the authors of the current manuscript are unaware of this published data set, since they fail to cite it.

The published paper already has established the importance for *klf*s in mediating very distinct lifespan-extending interventions, a body of evidence the current manuscript barely extends.

More importantly the published data convincingly establish *klf-3* as the most relevant factor. It is even there shown that overexpression of *klf-3* extends lifespan per se.

Hence and together, I feel that the current manuscript may become more suitable for a more specialized journal, provided that the associative dataset would be complemented by a number of experiments that establish causality for *klf-1*.

On a technical side note the authors may also wish to pay attention to reproducibility issues of their lifespan assays: As just one example, in the key Figure 1a N2 worms live up to 40 days while in other figures (1b, 5c, 6b) and consistent with another labs, N2s only live 30 days, indicating a 25% difference in lifespan within the authors' lab in N2 controls.

Reviewer #2 (Remarks to the Author):

Herholz et al. describe a novel role for *klf-1* in mediating a cytoprotective response in *C. elegans* through its regulation of cytochrome P450 oxidases. As these xenobiotic detoxification enzymes are known to be strongly linked to longevity, the authors show that *klf-1* is activated in mitochondrial mutant nematodes (and worms subjected to mitohormetic oxidative stress) and also required for their extended lifespan. These results provide interesting mechanistic insight into cytoprotective pathways modulating lifespan. Overall this is an exciting paper but there is one overarching concern and several major concerns in this reviewer's view.

Overarching concern: The authors have focused on KLF1. It is surprising that KLF3 is not discussed or studied at all. Of note, a recent paper in Nature Communications demonstrated a robust effect

of KLF1/3 double deficiency as conferring a robust effect on lifespan and that KLF3 overexpression extended lifespan. In light of this, the authors really need to assess the effect of KLF3 deficiency and dual-deficiency on xenobiotic detoxification enzymes. This prior work should also be cited.

Major concerns:

1. It is very intriguing that *klf-1* requirement for lifespan is only in adulthood and that knockdown of *klf-1* in the early developmental stages does not alter lifespan of mitochondrial mutant worms. While there may be technical challenges to measuring *klf-1* levels, the authors may consider the value of assessing degree of knockdown and how quickly *klf-1* transcript or protein levels drop, especially as RNAi of *klf-1* during the developmental stages likely does not occur immediately. If a *klf-1::GFP* reporter cannot be generated, perhaps RNAi against *klf-1* begun in parental worms, continued into the developmental stages of the progeny, then discontinued upon adulthood, could provide some confirmatory evidence that sufficient knockdown in the developmental period was achieved.

4. Figure 3 presents some rationale for a role for *klf-1* in mitohormesis, but not antioxidation. This reviewer is not convinced the protein carbonylation data completely exclude the possibility that *klf-1* alters antioxidation. Is the change in magnitude in WT worms from D1 to D5 smaller than the change in magnitude in WT worms subjected to RNAi *klf-1*? In other words, does depletion of *klf-1* increase age-related increase in protein carbonylation in WT worms and/or *isp-1;ctb-1* worms?

5. Fig 4 – The microarray is helpful. However, they do not speak to whether the various *cyp* genes are direct targets of *klf1*. The authors should consider ChIP studies....and preferably an unbiased Chip-Seq.

6. Figure 5 presents functional evidence of *klf-1* regulation of CYP functions and their relationship to lifespan. The claim that *klf-1* mediated lifespan modulation is dependent on CYP gene function is not well established. Does knockdown of *cyp* genes in a *klf-1* overexpressing worm alter lifespan, or does reintroduction of *cyp* in an *isp-1;ctb-1* worm subjected to RNAi *klf-1* rescue lifespan? In the same line of thought, in figure 7, can the transient block of ROS (and therefore block of activation of *klf-1*) by S3QEL-2 at L3 alter lifespan in *isp-1;ctb-1* mutants?

7. As referenced by the authors and in other mammalian studies, the KLFs (notably KLF4) regulate autophagy as well as steps in the mitochondrial life cycle. They should provide in Figure 6 evidence of changes in mito mass in worms at L4 and D1 in the context of RNAi *klf-1*. Also, some discussion of changes in gene expression of mitochondrial related genes in the D5 microarray could be useful, although an earlier timepoint would be preferred.

8. The generation of a *klf-1* OE worm is a powerful tool that is very much underutilized. Indeed, the *klf-1* OE lifespan extension is not impressive; however, as noted by the authors, nuclear localization of *klf-1* may be the primary determinant of activity. Therefore, have the authors treated *klf-1* OE worms with paraquat or crossed into the *isp-1;ctb-1* mutants and demonstrated changes in gene expression or function (lifespan, resistance to toxins, etc.)?

Minor concerns

Figure 3b seems to be missing a label for -/+ treatment with RNAi *klf-1*.

Statistical analysis of figure 3c-d seems inappropriate. Rather than T-test, tests accounting for multiple comparisons ought to be used.

In figure 2a, the effect of *klf-1* RNAi on *phsp-6::GFP* reporter strain could benefit from quantification, as the effect, as mentioned by the authors, is mild but seems real.

Studies have been published linking mammalian KLFs to mouse aging, and the authors utilize hep

cells to demonstrate conservation. Discussion could benefit from discussion of implications of liver xenobiotic detoxification mechanisms and mammalian lifespan.

Reviewer #3 (Remarks to the Author):

Studies in genetic model organisms like *C. elegans* have revolutionized our understanding of longevity. A few conditions/genetic regulatory paradigms have been discovered that extend lifespan in many animals, including mammals. The more difficult work has been pinpointing specific effector mechanisms that prolong lifespan and delay aging. Mutations in components of electron transport in mitochondria is one major paradigm for lifespan extension. The current study proposed a DNA regulatory protein named KLF-1 and downstream phase I detoxification genes as regulators of stress resistance (and lifespan) in worms with mutated mitochondria function. KLF proteins have previously been shown to regulate many age-related processes in mammals and one recent study showed that two *klf* genes promote lifespan under diverse conditions in *C. elegans*. The current study demonstrates that *klf-1* is required for longevity extension in *C. elegans* models of mitochondrial mutants. The authors identify phase I detoxification genes in the p450 family as downstream mechanisms and pinpoint late larval development as a critical time for activation of the pathway. KLF involvement in longevity is not completely novel, but it is topical and novel to the mitochondrial models. Detoxification in general is also not a new mechanism of longevity, but p450 and phase I is a new specific mechanism of importance to the general field of longevity. The studies appear to have generally been conducted in a logical manner and the conclusions are generally well supported and the findings are a good fit for the journal. However, there are a number of issues that need attention relating to putting the study into full context with prior literature, additional controls needed to support the conclusions, and reporting of the data (see below)

Major comments

Given that your model has *cyps* as the predominant longevity mechanism mitochondria and *klf-1*, overexpression of one or more of these *cyps* should increase lifespan. If not, then *cyps* are more likely one of many parallel or complementary mechanisms. Have the authors tested overexpression of *klf* or *cyps*? This should be included.

Many studies have previously implicated *klf* proteins in age-related processes and two *klf*s were recently shown to mediate lifespan in *C. elegans* (PMID: 29416266, 29030550). These should be incorporated in the introduction to put the current work into context.

KLF9 has been shown to regulate *CYP2D6* in mice (PMID:25217496). The authors should cite and incorporate this study into the discussion.

Autophagy was recently implicated as a longevity mechanism downstream from *klf-1/3*. How does this mechanism relate to the current proposed mechanism of phase I gene regulation? Did the authors conduct any experiments to investigate the role of autophagy in the mitochondria mutants? This needs to at least be discussed in the paper.

For all lifespan curves, the authors need to specify if one or all trials are plotted and data for individual trials needs to be listed in tables. This allows for evaluation of reproducibility.

Figure 1a - why is the longevity curve of the right so much shorter than the one on the left and in middle? Max is almost half on the right.

The microarray data needs to be submitted to a public source such as GEO Omnibus. This would obviate the need for posting the raw data here.

Minor comments

The title is generic to the point where it could be confused with *SKN-1/Nrf* for which there are numerous studies. I would recommend focusing or differentiating the title more.

Page 3 - There are three phases of detoxification including transport. Was phase III left out for a specific reason? Also, it's not accurate to say that little is known about regulation of xenophobic genes with respect to longevity given that these genes have been reported to be induced in many long lived conditions and under control of DAF-16, SKN-1, and others. Are the authors specifically referring to expression during aging, or something else? This statement needs to be more specific, or left out.

Page - "Cyp genes that encode enzymes that" is presumptuous. These are genes in the Cyp family, but you have not shown them to be bonafide functioning enzymes. This statement should be changed.

Figure 5 - Acrylamide is misspelled.

Page 16 - you have shown that klf has a conserved function in regulating phase 1 genes, but not longevity. This statement should be changed accordingly.

In the Figure legends, please specify what is meant by replicates. Are replicates different groups of worms from the same trial or separate trials? This needs to be clear for each panel. In some cases this is clear, but not all.

End of discussion - Many studies have implicated a complex mixture of effectors responsible for xenobiotic detoxification and other metabolic functions play a role in longevity in dietary restriction and insulin/IGF like mutants. This is not the first demonstration of this.

Supplemental tables - The tables are split oddly between pages with many pages blank and missing heading for columns. Perhaps this is a problem with conversion from a spreadsheet, but these tables are not useful in the current form and need to be reformatted. Posting to GEO Omnibus would obviate the need for these tables.

Methods - The calculation methods used for qPCR need to be described.

Point-by-point detailed answers to questions raised by the reviewers

We thank the reviewers for their helpful suggestions to improve our paper and their overall positive comments. Our response to the reviewers is as follows:

Reviewer #1:

Q1: As shown in Figure 7, the authors have generated a transgenic overexpressor of klf-1 under the control of an gut-specific constitutive promoter. The claim that klf-1 is a transcriptional regulator promoting longevity should be supported by demonstrating that such an overexpressor is long-lived per se.

We have already provided a lifespan analysis that shows moderate increase in longevity of KLF1 overexpressor (previous Fig 5c, now Supplementary Fig 6b). We showed that KLF1 activation (translocation to the nucleus) requires additional signal that in mitochondrial mutant is provided by increased ROS production. In the revised manuscript we provide evidence that knockdown of the two most highly upregulated *cyps* fully suppressed the observed upregulation of lifespan in the KLF1 overexpressor, further strengthening our results.

*Q2: Additionally and given the potential role of KLF-1 specifically in young-adult worms, the authors should generate a hsp-16.2 promoter which is gut-specific *and* heatshock-inducible to dissect the issue with developmental timing of expression and/or activity. The increased levels of ROS in N2s (as shown in this manuscript) should promote longevity in this background. Absence of such effect would fundamentally question the key claim of the manuscript.*

In our manuscript we showed that increased levels of ROS (paraquat) promote longevity in the N2 background (Fig 6a) and that this prolongation of the lifespan is dependent on KLF-1 presence, as the longevity could be fully suppressed when KLF-1 is depleted. Remarkably, as in the *isp-1; ctb-1* mutant, for the longevity induced by increased oxidative stress, the presence of KLF-1 is essential only during adulthood (Fig 6b-c). In addition, we have shown that *klf-1* expression is induced at D1 of adulthood in *isp-1; ctb-1* mutant and that suppression of KLF-1 levels at D1 is essential for the full suppression of the longevity (Supplementary Fig 1e).

In order to address the timing of KLF-1 activation we have performed experiments where we have evaluated the level of KLF-1 translocation into nucleus at different developmental stages. Our results demonstrated that KLF-1 is mainly present in the nucleus at L4 stage, to be quickly excluded from it upon transition to early adulthood (D1) in control animals. In the *isp-1;ctb-1* animals we observed

persistent KLF-1 localization at D1, in line with our previous analysis showing that D1 is crucial for the lifespan extension in mutants. Nuclear localization of KLF-1 is strongly prevented by the antioxidant vitamin C, providing further evidence that oxidative stress signalling plays important role in this process.

*Q3: Then, such overexpressors should be crossed into *skn-1(zu135)* mutants to prove the claimed independence from NRF2 signalling, to re-test the longevity phenotype.*

Since SKN-1 is essential for the worm development, *skn-1(zu135)* homozygotes do not produce viable offspring. Thus, genetic balancers are used to maintain the strain. Unfortunately, due to the time constraints of the revision process and despite our efforts, we were unable to obtain a properly balanced *klf-1* overexpressor strain in the *skn-1* mutant.

However, in our manuscript we have already presented strong data that SKN-1 is not involved in the longevity of *isp-1;ctb-1* mutant:

- the knockdown of *skn-1* does not significantly affect *isp-1;ctb-1* longevity (Supplementary Fig 6d).
- upon *skn-1* depletion, expression of *cyp-25a2* and *cyp-34a8* are either not affected, or even further increased in both N2 and *isp-1;ctb-1* mutant (Figs 5e and Supplementary Figure 6a).
- the expression of *gst-4*, *gcs-1* and catalases, bonafide SKN-1 targets was not affected or even decreased in *isp-1;ctb-1* mutant (Supplementary Fig 6c). In contrast, we show that KLF-1 does not regulate GST-4 levels (Supplementary Fig 6c).

*Q4: These findings could potentially extend findings published in Nature Communications in Oct. 2017 (Hsieh et al.) where all nematodal klf-genes have been studied in regards to their role in lifespan regulation, but also in a mammalian model. Apparently the authors of the current manuscript are unaware of this published data set, since they fail to cite it. The published paper already has established the importance for klfs in mediating very distinct lifespan-extending interventions, a body of evidence the current manuscript barely extends. More importantly the published data convincingly establish *klf-3* as the most relevant factor. It is even there shown that overexpression of *klf-3* extends lifespan per se. Hence and together, I feel that the current manuscript may become more suitable for a more specialized journal, provided that the associative dataset would be complemented by a number of experiments that establish causality for *klf-1**

We are truly sorry that we have missed to cite the manuscript by Hsieh et al. That manuscript indeed shows important contribution of high levels of KLF-3 to lifespan extension that seems to be regulating macroautophagy in worms. The authors also established connection of KLF-1 in this pathway, but its role seems to be less clear from the results present in the manuscript. Nevertheless, we strongly believe that our manuscript and the data present here unravel a different and very important role of KLF-1 in regulating longevity in mitochondrial mutants (previously not addressed by Hsieh et al.). We showed that KLF-1 regulates phase I detoxification pathway, primarily *cyp* genes and through this contributes to mitohormesis and increased resistance to oxidative stress and general increase in longevity. Therefore, this manuscript does not overlap with the manuscript Hsieh et al. and presents first strong evidence for the novel regulation of phase I detoxification, a pathway upregulated in all conditions that prolong lifespan in various species ranging from nematodes to mammals.

We have also extensively studied the role of KLF-3 in the context of KLF-1 regulation of Phase I detoxification pathway and these results could be found below in the reply to comments from Reviewer 2 (Q6). Additionally, we addressed the role of macroautophagy in *isp-1; ctb-1* mutants in our response to comments from reviewer 3(Q19). These results are now included in the Supplementary Fig 5.

Q5. On a technical side note the authors may also wish to pay attention to reproducibility issues of their lifespan assays: As just one example, in the key Figure 1a N2 worms live up to 40 days while in other figures (1b, 5c, 6b) and consistent with aother labs, N2s only live 30 days, indicating a 25% difference in lifespan within the authors' lab in N2 controls.

Indeed, in this particular experiment, both N2 and *isp-1; ctb-1* have shown unusually long lifespan, possibly due to fluctuations in temperature, as this was done in the different institute and with old equipment. We have now replaced the lifespan in question (Fig 1a) with other that indeed shows N2 maximal lifespan close to 30 days as shown by other studies.

Reviewer #2:

Herholz et al. describe a novel role for klf-1 in mediating a cytoprotective response in C. elegans through its regulation of cytochrome P450 oxidases. As these xenobiotic detoxification enzymes are known to be strongly linked to longevity, the authors show that klf-1 is activated in mitochondrial mutant nematodes (and worms subjected to mitohormetic oxidative stress) and also required for their extended lifespan. These results provide interesting mechanistic insight into cytoprotective pathways modulating lifespan. Overall this is an exciting paper but there is one overarching concern and several major concerns in this reviewer's view.

We thank the reviewer for the overall positive evaluation of our manuscript.

Q6: Overarching concern: The authors have focused on KLF1. It is surprising that KLF3 is not discussed or studied at all. Of note, a recent paper in Nature Communications demonstrated a robust effect of KLF1/3 double deficiency as conferring a robust effect on lifespan and that KLF3 overexpression extended lifespan. In light of this, the authors really need to assess the effect of KLF3 deficiency and dual-deficiency on xenobiotic detoxification enzymes. This prior work should also be cited.

We recognize our failure to cite the Hsieh et al. and have now included this in the revised manuscript. Furthermore, we have performed a series of experiments to address the potential role of KLF-3 in the context of xenobiotic detoxification and KLF-1. In the original manuscript we showed:

- that KLF-1 levels are upregulated in *isp-1; ctb-1* mutants at D1, while KLF-2 and KLF-3 are not changed (Supplementary Fig 5b)
- that depletion of KLF-3 levels actually promote further upregulation of certain *cyps*, in contrast to KLF-1 (Supplementary Fig 5a).

We have now strengthened these results by using *pcyp-25a1::rfp* reporter strain and show, KLF-3 depletion does not affect the expression of this gene and that combined *klf-1, klf-3* depletion leads to intermittent effect, possibly due to dilution of RNAi.

Our new results also show that KLF-3 strongly suppresses the lifespan of *isp-1; ctb-1* mutants, even below the N2 levels, suggesting an important role for this transcription factor in the context of mitochondrial dysfunction. However, this role is not related to regulation of phase I detoxification by KLF-1.

Remarkably, depletion of both *klf-1*, *klf-3* also lead to only moderate suppression of longevity in *isp-1;ctb-1* mutants, again, likely due to dilution of both RNAi arguing against possible role for macroautophagy in mitochondrial mutants, as Hsieh et al. showed that depletion of both KLF-1 and KLF-3 has a synergistic effect on the autophagy in *daf-2* and *eat-2* mutants.

These results are now incorporated in Supplementary Figure 5.

Q7 - It is very intriguing that klf-1 requirement for lifespan is only in adulthood and that knockdown of klf-1 in the early developmental stages does not alter lifespan of mitochondrial mutant worms. While there may be technical challenges to measuring klf-1 levels, the authors may consider the value of assessing degree of knockdown and how quickly klf-1 transcript or protein levels drop, especially as RNAi of klf-1 during the developmental stages likely does not occur immediately. If a klf-1::GFP reporter cannot be generated, perhaps RNAi against klf-1 begun in parental worms, continued into the developmental stages of the progeny, then discontinued upon adulthood, could provide some confirmatory evidence that sufficient knockdown in the developmental period was achieved.

We agree with the reviewer that this is a very important question. We have now performed the suggested experiment and kept knockdown of *klf-1* over two generations (parental worms on *klf-1* the whole lifespan and progeny only until adulthood) and obtained the same result as previously – KLF-1 is needed for the lifespan prolongation during adulthood and is dispensable during development. These results are now incorporated in Supplementary Figure 1.

Q8 - 4. Figure 3 presents some rationale for a role for klf-1 in mitohormesis, but not antioxidation. This reviewer is not convinced the protein carbonylation data completely exclude the possibility that klf-1 alters antioxidation. Is the change in magnitude in WT worms from D1 to D5 smaller than the change in magnitude in WT worms subjected to RNAi klf-1? In other words, does depletion of klf-1 increase age-related increase in protein carbonylation in WT worms and/or isp-1;ctb-1 worms?

The protein carbonylation indeed increased in control N2 worms from D1 to D5 and although, at D1 we did not observe that this is regulated by KLF-1, at D5 levels of carbonyls increased even more when we depleted KLF-1. As correctly noticed by the reviewer the difference is even more significant ($n > 0,001$) when we compared control N2 D1 worms with D5 N2 worms upon *klf-1* RNAi. We are sorry that this was not put in the original manuscript and have now corrected this mistake.

Initially we were also surprised by the fact that KLF-1 does not directly regulate antioxidant response. However, several lines of evidence point into this direction:

- We have not detected changes in antioxidant enzyme expression in our microarray data and they were not regulated by KLF-1
- We have not detected changes in the expression of different catalases in *isp-1;ctb-1* mutants
- Observed changes in the expression of SODs (beside maybe *sod-3* know to be one of the primary targets of DAF-16) were not consistent with the direct KLF-1 regulation

Therefore, we strongly believe that the antioxidant response is not directly regulated by KLF-1 and that this regulation is, like UPRmt and HIF-1a response, secondary to the KLF-1 depletion, regulated by some other transcription factor, possibly SKN-1 or DAF-16.

Q9 -5. Fig 4 – The microarray is helpful. However, they do not speak to whether the various cyp genes are direct targets of klf1. The authors should consider ChIP studies....and preferably an unbiased Chip-Seq.

Indeed, CHIPSeq analysis would be very helpful in showing direct binding of KLF-1 to promoter regions of direct target genes. Unfortunately, we do not have an antibody against KLF-1 and our preliminary pull-down assay using anti-GFP antibody in the strain expressing GFP tagged KLF-1 followed by MS analysis was quite disappointing as the KLF-1 was not detected among first 30 proteins. Discouraged by this result, we decided against doing CHIPSeq analysis. As an alternative, we have analysed upstream promoters of different *cyp* genes and all microarray targets presented in Fig 4a. We detected classical CA/GCCC sequence in many of the analysed genes, suggesting that, indeed they are direct KLF-1 targets. These results are now presented as the Supplementary Table 3.

Q10 -6. Figure 5 presents functional evidence of klf-1 regulation of CYP functions and their relationship to lifespan. The claim that klf-1 mediated lifespan modulation is dependent on CYP gene function is not well established. Does knockdown of cyp genes in a klf-1 overexpressing worm alter lifespan, or does reintroduction of cyp in an isp-1;ctb-1 worm subjected to RNAi klf-1 rescue lifespan?

We thank the reviewer for raising this very important question. We have depleted *cyps* in *klf-1* overexpressing strain and indeed been able to show that this intervention suppresses a moderate increase in lifespan.

We also tried to reintroduce *cyps* into *isp-1; ctb-1* mutants, but this strongly affected their development. This might be the consequence of various effects, not least of them energy crisis caused by low levels of ATP in the *isp-1; ctb-1* mutants combined with highly energy demanding CYPs. Remarkably, knockdown of either *cyp-25A1* or *cyp13A11* partially normalized the developmental delay in *isp-1; ctb-1* mutants.

This effect was not observed after *klf-1* knockdown, suggesting two different possibilities: (i) *klf-1* knockdown suppressed *cyp* levels to around 80% while direct *cyp* RNAi almost completely depleted them (99%) and therefore we could not observe the same effect on development; or (ii) additional mechanisms might be present that counteract the beneficial effect of lower CYP levels during worm development. Nevertheless, this is an example of time-sequential antagonistic pleiotropy in which the same gene has beneficial net value at one stage in an animal's life and adverse net consequences at another stage. As we do not fully understand this effect yet and this was not the focus of current manuscript, we decided not to include these results in the revised version.

Q11 - In the same line of thought, in figure 7, can the transient block of ROS (and therefore block of activation of klf-1) by S3QEL-2 at L3 alter lifespan in isp-1;ctb-1 mutants?

This is another very important question that we addressed in our revised manuscript. Unfortunately, we found S3QEL-2 to be quite unreliable for the long lasting lifespan analysis as it has to be administered as drops on top of the bacterial lawn and worms seem to avoid it. Therefore, for the lifespan studies, we used Vitamin C which we could put directly into agar plates. We could show that blockage of ROS production from L3 fully suppresses the longevity signal, while suppression from D1 only partially suppresses longevity *isp-1; ctb-1* longevity, strongly indicating that signal that promotes KLF-1 activation largely depends on the L3-young adulthood transition.

The fact that period from L3 to early adulthood (D1) is important for the activation of protective stress responses in *isp-1;ctb-1* mutant is further supported by our finding that treatment with antimycin A, a potent inducer of ROS production at complex III, during this specific period paradoxically provides strong resistance to oxidative stress (Fig.7e).

Conversely, treatment with S3QEL-2 during the same period (L3-D1) strongly decreased H₂O₂ resistance in *isp-1;ctb-1* mutants. While treatment throughout development, ending at L4 had milder effect, that was not statistically different.

Correspondingly, Vitamin C treatment also decreases expression of *cyp-25a2* reporter when administered during the same period.

Q12 -7. As referenced by the authors and in other mammalian studies, the KLFs (notably KLF4) regulate autophagy as well as steps in the mitochondrial life cycle. They should provide in Figure 6 evidence of changes in mito mass in worms at L4 and D1 in the context of RNAi klf-1. Also, some discussion of changes in gene expression of mitochondrial related genes in the D5 microarray could be useful, although an earlier timepoint would be preferred.

We thank the reviewer for raising this important question. We have analysed changes in mitomass, this time in context of KLF-1 role in this process. We again showed that mitochondrial mass increases from L4 to D1 and this increase is higher in *isp-1; ctb-1* mutants but not dependent on KLF-1.

We have included discussion about mitochondrial gene expression changes in our microarray data. We also performed W. blots to evaluate changes in the level of OXPHOS subunits in control and mutant worms at D1 upon depletion of KLF-1, KLF-3 or both and did not detect any changes as shown below for analysis of NUO-2 levels (NDUFS3 homolog).

Q13 - 8. The generation of a klf-1 OE worm is a powerful tool that is very much underutilized. Indeed, the klf-1 OE lifespan extension is not impressive; however, as noted by the authors, nuclear localization of klf-1 may be the primary determinant of activity. Therefore, have the authors treated klf-1 OE worms with paraquat or crossed into the isp-1; ctb-1 mutants and demonstrated changes in gene expression or function (lifespan, resistance to toxins, etc.)?

As reviewer pointed out, we think that moderate increase in the lifespan in comparison to *isp-1, ctb-1* mutant is likely due to the fact that additional signal is needed for KLF-1 to translocate to the nucleus and activate its genes. As suggested, we have now performed additional experiments to show that *klf-1* overexpressors are indeed more resistant to treatment with high levels of H₂O₂. Importantly, this resistance is fully dependent on CYPs in both control and *klf-1* overexpressor worms. This fully complements results from the lifespan analysis where knockdown of *cyps* suppresses lifespan extension mediated by *klf-1* overexpression (see comment to Q4).

Q14 - Figure 3b seems to be missing a label for +/- treatment with RNAi klf-1. Statistical analysis of figure 3c-d seems inappropriate. Rather than T-test, tests accounting for multiple comparisons ought to be used.

This obvious mistake from our side has been now corrected in the revised manuscript. We have also used OneWay Anova to analyse our results and this is now included in the revised manuscript.

Q15 - In figure 2a, the effect of klf-1 RNAi on hsp-6::GFP reporter strain could benefit from quantification, as the effect, as mentioned by the authors, is mild but seems real.

Quantification of results in Figure 2a is now included in the revised manuscript. The quantification analysis showed that there is no significant difference in hsp-6 activation upon KLF-1 knockdown.

Q16 - Studies have been published linking mammalian KLFs to mouse aging, and the authors utilize hep cells to demonstrate conservation. Discussion could benefit from discussion of implications of liver xenobiotic detoxification mechanisms and mammalian lifespan.

We have now included additional part in the discussion about mammalian ageing and xenobiotic detoxification response in this process.

Reviewer #3:

Studies in genetic model organisms like C. elegans have revolutionized our understanding of longevity. A few conditions/genetic regulatory paradigms have been discovered that extend lifespan in many animals, including mammals. The more difficult work has been pinpointing specific effector mechanisms that prolong lifespan and delay aging. Mutations in components of electron transport in mitochondria is one major paradigm for lifespan extension. The current study proposed a DNA regulatory protein named KLF-1 and downstream phase I detoxification genes as regulators of stress resistance (and lifespan) in worms with mutated mitochondria function. KLF proteins have previously been shown to regulate many age-related processes in mammals and one recent study showed that two klf genes promote lifespan under diverse conditions in C. elegans. The current study demonstrates that klf-1 is required for longevity extension in C. elegans models of mitochondrial mutants. The authors identify phase I detoxification genes in the p450 family as downstream mechanisms and pinpoint late larval development as a critical time for activation of the pathway. KLF involvement in longevity is not completely novel, but it is topical and novel to the mitochondrial models. Detoxification in general is also not a new mechanism of longevity, but p450 and phase I is a new specific mechanism of importance to the general field of longevity. The studies appear to have generally been conducted in a logical manner and the conclusions are generally well supported and the findings are a good fit for the journal. However, there are a number of issues that need attention relating to putting the study into full context with prior literature, additional controls needed to support the conclusions, and reporting of the data.

We thank the reviewer for the overall positive evaluation of our manuscript.

Major comments:

Q17 - Given that your model has cyps as the predominant longevity mechanism mitochondria and klf-1, overexpression of one or more of these cyps should increase lifespan. If not, then cyps are more likely one of many parallel or complementary mechanisms. Have the authors tested overexpression of klf or cyps? This should be included.

Indeed, this is an important point that has been raised by all three reviewers and therefore, in the revised manuscript we have now included the lifespan analysis of *klf-1* overexpressor that indeed moderately increase lifespan (Fig 5d). In addition, we demonstrated that knockdown of the two most highly upregulated cyps fully suppressed the observed upregulation of lifespan in the overexpressor, strengthening our previous results.

As many cyps seems to be overexpressed in *isp-1;ctb-1* mutant, many of them regulated by KLF-1, we were not sure if overexpressing only one of them in worms would affect longevity. Given the short time that we had for the revision of our manuscript and the number of new analysis we did, with heavy heart we decided against testing if overexpression of different cyps would influence the lifespan in hope that the reviewer will understand our reasoning in this case.

Q18 - Many studies have previously implicated klf proteins in age-related processes and two klfs were recently shown to mediate lifespan in C. elegans (PMID: 29416266, 29030550). These should be incorporated in the introduction to put the current work into context. KLF9 has been shown to regulate CYP2D6 in mice (PMID:25217496). The authors should cite and incorporate this study into the discussion.

We recognize our mistake of not incorporating these important papers in the original manuscript and have now added paragraphs in the discussion of our revised manuscript where we cited all suggested manuscripts.

Q19 - Autophagy was recently implicated as a longevity mechanism downstream from klf-1/3. How does this mechanism relate to the current proposed mechanism of phase I gene regulation? Did the authors conduct any experiments to investigate the role of autophagy in the mitochondria mutants? This needs to at least be discussed in the paper.

Indeed, a role for the KLF-1 has been suggested by the recent paper by Hsieh et al. and we have now performed a number of experiments to address further the role of macroautophagy in the context of mitochondrial mutant and KLF-1. First of all, we look at the expression of most important proteins involved in the formation of autophagosome and did not observed any consistent change. In general, they show normal, or even lower level of expression in *isp-1;ctb-1* mutants and are not regulated by KLF-1. This is a very important point, as at D1 we detected high upregulation of KLF-1 levels in *isp-1;ctb-1* that are reflected in massive upregulation of cyp transcripts (Fig 4b), while autophagy genes show tendencies for lower average levels.

Next, using a GFP::LGG-1 translational reporter, we have not seen any change in the number of GFP-positive puncta in hypodermal seam cells, often used as indicator of autophagy, and their level did not change whether we depleted *klf-1*, *klf-3* or both factors in the same time.

We also measured the autophagy flux (cleavage of GFP::LGG-1 inside lysosome) to determine the contribution of KLF-1 and KLF-3 to this process. The autophagy flux did not change in *isp-1; ctb-1* mutants, while depletion of KLF1 seem to moderately increased it in this background. The effect is further exaggerated by addition of Vitamin C, suggesting that KLF-1 and its activation by mitochondrial ROS might play a role in suppression and not activation of autophagy.

Taken together, these results argue against major role for the autophagy in *isp-1; ctb-1* mutants and question proposed role for the KLF-1 in this process. These results are now added to Supplementary Figure 5.

Q20 - For all lifespan curves, the authors need to specify if one or all trials are plotted and

data for individual trials needs to be listed in tables. This allows for evaluation of reproducibility.

All lifespans presented in the manuscript have been independently repeated at least twice and this data is included in the Supplementary Table 4.

Q21 - Figure 1a - why is the longevity curve of the right so much shorter than the one on the left and in middle? Max is almost half on the right.

Indeed, in this particular experiment, both N2 and *isp-1; ctb-1* have shown unusually long lifespan, possibly due to fluctuations in temperature, as this was done in the different institute and with old equipment. We have now replaced the lifespan in question (Fig 1a) with other that shows N2 maximal lifespan close to 30 days and *isp-1; ctb-1* maximal lifespan close to 40 days.

Q22 - The microarray data needs to be submitted to a public source such as GEO Omnibus. This would obviate the need for posting the raw data here.

We have now replaced the raw microarray data with the GEO links in the revised manuscript.

Minor comment

Q23 - The title is generic to the point where it could be confused with SKN-1/Nrf for which there are numerous studies. I would recommend focusing or differentiating the title more.

As suggested by the reviewer we have now changed the title of our manuscript to: “KLF-1 orchestrates a xenobiotic detoxification program essential for longevity of mitochondrial mutants”

Q24 - Page 3 - There are three phases of detoxification including transport. Was phase III left out for a specific reason? Also, it's not accurate to say that little is known about regulation of xenophobic genes with respect to longevity given that these genes have been reported to be induced in many long lived conditions and under control of DAF-16, SKN-1, and others. Are the authors specifically referring to expression during aging, or something else? This statement needs to be more specific, or left out.

We have now changed the sentence “little is currently known about the genes and pathways underlying these cytoprotective responses and their ability to affect lifespan.” to “little is currently

known about signalling pathways underlying these cytoprotective responses and their ability to affect lifespan.”. We have also included a description of Phase III detoxification in our introduction: “. . .and phase III transporters that mediate the efflux of metabolic end products out of the cells after the completion of Phase II conjugation”.

Q25 -Page - “Cyp genes that encode enzymes that” is presumptuous. These are genes in the Cyp family, but you have not shown them to be bonafide functioning enzymes. This statement should be changed.

We have now changed this statement in the manuscript to “cyp genes that in different organisms often encode enzymes”

Q26 - Figure 5 - Acrylamide is misspelled.

This has been corrected.

Q27 - Page 16 - you have shown that klf has a conserved function in regulating phase I genes, but not longevity. This statement should be changed accordingly.

This statement has been changed accordingly

Q28 - In the Figure legends, please specify what is meant by replicates. Are replicates different groups of worms from the same trial or separate trials? This needs to be clear for each panel. In some cases this is clear, but not all.

We have now clarified this in all figure legends.

Q29 - End of discussion - Many studies have implicated a complex mixture of effectors responsible for xenobiotic detoxification and other metabolic functions play a role in longevity in dietary restriction and insulin/IGF like mutants. This is not the first demonstration of this.

We have changed this statement in the revised manuscript

Q30 - Supplemental tables - The tables are split oddly between pages with many pages blank and missing heading for columns. Perhaps this is a problem with conversion from a spreadsheet, but these tables are not useful in the current form and need to be reformatted. Posting to GEO Omnibus would obviate the need for these tables.

Indeed, the automatic conversion of Excel files into PDFs likely caused this strange appearance in the submitted manuscript. We have now replaced them with GEO submission numbers.

Q31 - Methods - The calculation methods used for qPCR need to be described.

We have included the calculation method used for qPCR in the revised manuscript.

We thank you for considering our revised manuscript.

Reviewers' comments:

Reviewer #1 (Remarks to the Author):

The reviewer has no additional comments.

Reviewer #2 (Remarks to the Author):

The revised manuscript is improved, but there are important considerations listed below that really need to be resolved before supporting the publication of this work.

1. The findings that autophagy gene expression and autophagy flux by the well known GFP::LGG-1 translational reporter are not altered significantly in the *isp-1;ctb-1* nematode are interesting. Logically, enhancement of autophagy in the *isp-1;ctb-1* animal will be additive with the activation of xenobiotic detoxification pathways in extending lifespan. The authors should demonstrate this dispositive result to exclude the contribution of autophagy.

2. We appreciate the difficulty in obtaining antibodies of ChIP quality as well as the difficulty in pulldown of tagged proteins (e.g. GFP tagged KLF-1). The challenge of antibodies is quite common but frankly can be easily overcome using properly tagged constructs. In general, the use of a FLAGx3 or His tagged proteins rather than bulky fluorescent tags works more optimally. We would strongly encourage the authors to perform these type of studies because the presence of C(A/G)CCC sequences upstream of target genes is helpful but far from convincing. The reality is that such sequences are common and their presence does little to prove that they are a direct target. The most convincing evidence is to do unbiased ChIP-seq studies and then utilize that information to perform promoter mutation studies in gene reporter-luciferase studies. These crucial studies will convincingly demonstrate that KLF-1 is recruited to promoters as a direct transcriptional regulator, rather than through a secondary mediator.

3. The authors show that *klf-1* overexpressing worms are more resistant to high levels of H₂O₂; this is consistent with the claims. If corresponding observations could be seen in *isp-1;ctb-1* worms overexpressing *klf-1* (admittedly with the loss of temporal control of *klf-1* expression), that would be convincing.

4. It is very intriguing that *cyp* reintroduction into *isp-1;ctb-1* mutants affected development and that this effect did not occur with RNAi of *klf-1*. The authors suggest that extended development may be due to low levels of ATP coupled with highly energy demanding *cyps*. We agree that this is not a central focus of the current work. Still, the authors should provide a measurement of ATP levels in the worm at this crucial period of *cyp* gene expression, as it would be expected to correlate with *cyp* activity.

Reviewer #3 (Remarks to the Author):

My comments have been addressed adequately. Thank you.

Point-by-point detailed answers to questions raised by the reviewer

We thank the reviewers 1 and 3 for their affirmative evaluation of our manuscript. We also thank reviewer 2 for the additional, helpful suggestions to improve our paper. We believe that in the second revision we have addressed all his concerns in full.

*1. The findings that autophagy gene expression and autophagy flux by the well known GFP::LGG-1 translational reporter are not altered significantly in the *isp-1;ctb-1* nematode are interesting. Logically, enhancement of autophagy in the *isp-1;ctb-1* animal will be additive with the activation of xenobiotic detoxification pathways in extending lifespan. The authors should demonstrate this dispositive result to exclude the contribution of autophagy.*

Autophagy is an essential process in all organisms even in wild type state. This has been shown also for *C. elegans* in multiple occasions, most recently in the latest paper from Suokas lab (PMID: 30929899), in which they showed that knockdown of either *lgg-1* or *bec-1* significantly decreases lifespan of N2 animals. Similarly, it has also been shown that autophagy is important for multiple longevity pathways including mitochondrial mutants (PMID: 23925298). We have also provided strong evidence that in *isp-1; ctb-1* mutant, this process is not regulated by KLF-1. This data is further strengthened by results of our ChIP-Seq analysis in which we did not find any of the genes involved in autophagy, to be direct KLF-1 targets. Therefore, we believe that dissecting further involvement of autophagy in this particular pathway would not shed any additional light on the mechanism of KLF-1 action and would rather be inconclusive.

2. We appreciate the difficulty in obtaining antibodies of ChIP quality as well as the difficulty in pull-down of tagged proteins (e.g. GFP tagged KLF-1). The challenge of antibodies is quite common but frankly can be easily overcome using properly tagged constructs. In general, the use of a FLAGx3 or His tagged proteins rather than bulky fluorescent tags works more optimally. We would strongly encourage the authors to perform these type of studies because the presence of C(A/G)CCC sequences upstream of target genes is helpful but far from convincing. The reality is that such sequences are common and their presence does little to prove that they are a direct target. The most convincing evidence is to do unbiased ChIP-seq studies and then utilize that information to perform promoter mutation studies in gene reporter-luciferase studies. These crucial studies will convincingly demonstrate that KLF-1 is recruited to promoters as a direct transcriptional regulator, rather than through a secondary mediator.

As suggested by reviewer and editor, we have now put additional effort and successfully performed ChIP-Seq analysis of KLF-1 targets. This time we used a GFP-Trap (Chromotek) to pull down GFP tagged KLF-1 protein. The successful KLF-1 pull-down was verified using mass spectrometry.

Chromatin immunoprecipitation followed by parallel DNA sequencing (ChIP-seq) was in N2 overexpressing KLF-1-GFP protein or *isp-1;ctb-1* worms. The analysis identified a number of genomic regions (on average 430 and 550 targets, respectively (n=3)), out of which 148 were found in both data sets (Supplementary Table 4 in the revised manuscript). We identified a KLF-1-bound region immediately upstream of the 4 *cyp*, including *cyp-13A11*, one of the most upregulated genes in our microarray data set. Further qPCR analysis on ChIP samples demonstrated that, indeed KLF-1 is present on *cyp-13A11* promoter region in N2 worms, but it binds the

same region in *isp-1;ctb-1* mutants much more efficiently (Fig. 4c in the revised manuscript). Remarkably, the ChIP-Seq data not only strongly overlapped with our microarray results, but more than 80% of detected KLF-1 targets was in genes previously shown to be responsive to either, treatment with oxidative-stress producing agents (paraquat or rotenone), or agents that block mtDNA synthesis and therefore lead to strong mitochondrial OXPHOS dysfunction (EtBr or nucleoside reverse transcriptase inhibitors –NRTIs), and in most cases KLF-1 target genes were found in more than one data set.

We would have hoped for more target genes identified with ChIP-Seq analysis, but this result reflects well known difficulty of the isolation of chromatin from worms, combined with the limitation imposed by the lack of antibodies against endogenous genes. Still, our data strongly support our previous conclusions that KLF-1 indeed directly regulates expression of *cyp* genes.

3. *The authors show that klf-1 overexpressing worms are more resistant to high levels of H2O2; this is consistent with the claims. If corresponding observations could be seen in isp-1;ctb-1 worms overexpressing klf-1 (admittedly with the loss of temporal control of klf-1 expression), that would be convincing.*

We have already shown in our revised manuscript that, both *isp-1; ctp-1* (already have upregulated KLF-1 levels), and worms overexpressing KLF-1, have increased resistance to oxidative stress (Fig 5c and Supplementary Fig 7f, respectively). It is unclear if further overexpression of KLF-1 in *isp-1, ctb-1* background would affect this phenotype or provide any additional insight into this process.

4. *It is very intriguing that cyp reintroduction into isp-1;ctb-1 mutants affected development and that this effect did not occur with RNAi of klf-1. The authors suggest that extended development may be due to low levels of ATP coupled with highly energy demanding cyps. We agree that this is not a central focus of the current work. Still, the authors should provide a measurement of ATP levels in the worm at this crucial period of cyp gene expression, as it would be expected to correlate with cyp activity.*

Overexpression of *cyp(s)* has indeed detrimental effect on *isp-1, ctb-1* mutant, as it extremely prolongs and stalls their development. Opposite is true when we deplete *cyp-13A11* and *cyp-25A1*, we actually correct the developmental defect observed in *isp-1, ctb-1* mutant.

The *klf-1* RNAi would not the same effect as overexpression of *cyps*, but exactly the opposite, it should mimic depletion of *cyps*. Remarkably, *klf-1* RNAi does not have an effect on *isp-1, ctb-1* development, likely because direct depletion of

cyps has a much stronger effect on their levels than *klf-1* RNAi. As pointed out before, we strongly believe that this is outside of the scope of this manuscript and are still working to shed more light on this important question.

We thank you for considering our revised manuscript.

REVIEWERS' COMMENTS:

Reviewer #2 (Remarks to the Author):

The authors have answered my lingering concerns.

Reviewer #2:

The authors have answered my lingering concerns.

We thank the reviewer 2 for hers/his affirmative evaluation of our manuscript.